# Inexact Augmented Lagrangian Methods for Conic Programs: Quadratic Growth and Linear Convergence

**Feng-Yi Liao**[1]    **Lijun Ding**[2]    **Yang Zheng**[1] *

[1]Department of Electrical and Computer Engineering, UC San Diego
[2]Department of Mathematics, UC San Diego

## Abstract

Augmented Lagrangian Methods (ALMs) are widely employed in solving constrained optimizations, and some efficient solvers are developed based on this framework. Under the quadratic growth assumption, it is known that the dual iterates and the Karush–Kuhn–Tucker (KKT) residuals of ALMs applied to semidefinite programs (SDPs) converge linearly. In contrast, the convergence rate of the primal iterates has remained elusive. In this paper, we resolve this challenge by establishing new *quadratic growth* and *error bound* properties for primal and dual SDPs under the strict complementarity condition. Our main results reveal that both primal and dual iterates of the ALMs converge linearly contingent solely upon the assumption of strict complementarity and a bounded solution set. This finding provides a positive answer to an open question regarding the asymptotically linear convergence of the primal iterates of ALMs applied to semidefinite optimization.

## 1 Introduction

We consider the standard primal and dual semidefinite programs (SDPs) of the form

$$
\begin{aligned}
p^\star := \min_{X \in \mathbb{S}^n} \quad & \langle C, X \rangle \\
\text{subject to} \quad & \mathcal{A}(X) = b, \\
& X \in \mathbb{S}^n_+,
\end{aligned}
\qquad (\text{P})
\qquad
\begin{aligned}
d^\star := \max_{y \in \mathbb{R}^m, Z \in \mathbb{S}^n} \quad & \langle b, y \rangle \\
\text{subject to} \quad & \mathcal{A}^*(y) + Z = C, \\
& Z \in \mathbb{S}^n_+,
\end{aligned}
\qquad (\text{D})
$$

where the problem data consists of a cost matrix $C \in \mathbb{S}^n$, a constant vector $b \in \mathbb{R}^m$, a linear map $\mathcal{A} : \mathbb{S}^n \to \mathbb{R}^m$ $\mathcal{A}(X) = [\langle A_1, X \rangle, \cdots, \langle A_m, X \rangle]^\mathsf{T}$ with $A_1, \cdots, A_m \in \mathbb{S}^n$, and its adjoint operator $\mathcal{A}^* : \mathbb{R}^m \to \mathbb{S}^n$ as $\mathcal{A}^*(y) = \sum_{i=1}^m A_i y_i$ that satisfy $\langle \mathcal{A}(X), y \rangle = \langle X, \mathcal{A}^*(y) \rangle, \forall X \in \mathbb{S}^n, y \in \mathbb{R}^m$.

SDPs are a broad and powerful class of conic programs, which include linear programs and second-order cone programs as special cases. The class of SDPs has found an extensive list of applications, including control theory [1], combinatorial optimization [2], polynomial optimization [3], machine learning [4–6], and beyond [7]. Meanwhile, many algorithms have been developed to solve (P) and (D), ranging from reliable interior point methods [8–10] to scalable first-order methods [11–14]. In particular, it is demonstrated that Augmented Lagrangian methods (ALMs) [15, 16] are suitable for solving large-scale optimization problems. Some efficient ALM-based algorithms have been developed to solve (P) and (D). For example, the celebrated low-rank matrix factorization [17] was integrated with ALM for efficient algorithm design. A semi-smooth Newton-CG algorithm was proposed for ALM to solve SDPs with many affine constraints in [18]. An enhanced version was introduced in [19] to tackle degenerate SDPs by combining a warm starting strategy. The algorithms [18, 19] have been implemented into a solver, SDPNAL+, which has shown very impressive numerical performance in benchmark problems. Other recent ALM-based algorithms include [20–22].

---

*Corresponding author

Despite the impressive practical performance, theoretical convergence guarantees of applying ALMs to (P) and (D) are less complete. Thanks to the seminal work [23], it is well-known that running ALMs for the primal (P) is equivalent to executing the proximal point method (PPM) for the dual (D). Thus, the classical convergences of ALM are typically inherited from PPM, which only guarantees that the dual iterates converge linearly when a *Lipschitz* continuity property is satisfied [24]. The Lipchitz condition requires (P) to have a unique solution. The uniqueness assumption was relaxed in [25], which allows multiple optimal solutions but requires an *error bound* property. Recently, this error bound property was established in [26] in a general conic optimization setup. The work [26] also proves linear convergence of KKT residuals when applying ALM to convex composite conic programming. However, due to the connections between ALM and PPM, all these results only guarantee linear convergence for the dual iterates, while the convergence rate of the primal iterates in ALM remains unclear. Indeed, it is noted in [21, 26] that the problem of *whether the primal sequence generated by the ALM can also converge asymptotically linearly* is open.

This asymmetry in convergence guarantees is notably dissatisfying, especially considering the elegant duality in (P) and (D). In this paper, our main goal is to establish linear convergence guarantees for both primal and dual iterates when applying an inexact ALM to SDPs (either (P) or (D)), under the usual assumption of strong duality and strict complementarity.

**Our contributions.** To resolve the challenge, we have made novel contributions from two aspects. On the problem structure side:

(a) Under the standard *strict complementarity* assumption, we establish the *quadratic growth* and *error bound* for both *primal and dual* SDPs (P) and (D) over any compact set containing an optimal solution (see Theorems 1 and 2). Our results not only extend the findings of [27] from a small (yet unknown) neighborhood to any compact set, benefiting the analysis of iterative algorithms, but also improves the recent results in [28–30]; see Remark 1 for a comparison.
(b) To establish the *quadratic growth*, we unveil a *new characterization* of the preimage of the subdifferential of the exact penalty functions for SDPs. This reveals a useful connection between exact penalty functions and indicator functions (see (11) in Lemma 3). We believe that this new characterization is of independent interest.
(c) Using the new characterization, we further provide a new and simple proof for the growth properties in the exact penalty functions of $\mathbb{S}_+^n$ (see Appendix B), and clarify some subtle differences in constructing exact penalty functions (see Section 3.1 and Remark B.1).

On the algorithm analysis side:

(a) By leveraging the established quadratic growth and error bound, we show that applying ALM to solve either the primal or dual SDPs (P) and (D) has linear convergence for both the primal and dual iterates (see Theorem 3).
(b) We establish symmetric versions of inexact ALMs for solving both the primal and dual conic programs (see Section 4 and Appendix E.3), which is less emphasized in the literature.
(c) We clarify the subtlety concerning the (in)feasibility of the dual iterates by inexact ALMs (see Section 4.2), which is an important issue concerning the relationship between PPM and ALM.

**Related works**: We here review some closely related results in the literature.

**Quadratic growth and error bound.** Many studies have been dedicated to establishing *quadratic growth* or *error bound* for conic optimizations under various assumptions [28, 31, 27, 32–35]. Starting from the famous Hoffman's lemma [36], a global error bound for linear systems has been established, which was later extended to a partially infinite-dimensional case [37]. Sturm investigated the error bound for SDPs, in which the exponent term is bounded by $2^d$ with $d$ being the *sigularity* degree that is at most $n - 1$ [32]. Zhang provided a simple proof of error bound for a feasibility system under Slater's condition [33]. Zhou and So provided a unified proof to establish error bound for a class of structured convex optimization [35]. The results in [35] are generalized to convex problems with spectral functions [31]. Drusvyatskiy and Lewis discussed quadratic growth for a convex problem of the form $\min_x f(Ax) + g(x)$ under *dual strict complementarity* [34, Section 4]. Very recently, Ding and Udell established an error bound property for general conic programs via an elementary framework using *strict complementarity* [28]. All the aforementioned results have slightly different assumptions, resulting in different forms of quadratic growth properties. They are not directly applicable to our purpose of establishing linear convergences of primal and dual iterates in ALM; a detailed comparison will be provided in Remark 1 after introducing our main result.

**Augmented Lagrangian method (ALM) and Proximal point method (PPM).** The ALM is first introduced in [15, 16] to improve numerical performance of penalty methods. It is shown by Rockafellar that the augmented dual function is continuously differentiable [38], and the ALM can be viewed as an augmented dual ascent method [24], thus the convergence of ALM can be analyzed via dual ascent. The seminal work [23] established a strong relation between ALM and PPM: the dual iterate by ALM is the same as the proximal update on the dual function by PPM. From the convergence of PPM, we know that the dual iterates of ALM converge linearly when the preimage of the subdifferential map of the dual function is *Lipschitz* continuous at the origin [24]. However, the Lipchitz condition requires the solution to be unique, which is not suitable for many practical applications. Luque made an important extension in PPM to allow multiple solutions while maintaining linear convergence [25]. Recently, Cui et. al relaxed the Lipschitz assumption to *upper*-Lipschtiz continuous which allows multiple primal and dual solutions [26]. In addition, the KKT residuals of convex composite conic programs are shown to converge linearly under the assumption that the dual function satisfies quadratic growth [26]. The primal iterates of ALM are also known to converge linearly under a much stronger Lipschitz assumption on the augmented Lagrangian function [23, 26]. As pointed out in [26, Section 3], the Lipschitz property of the Lagrangian function is more conservative than the quadratic growth of the primal or dual function. To the best of our knowledge, when applying ALM to conic optimization (P) and (D), the problem of whether the primal iterates converge linearly under the normal quadratic growth condition remains open.

**Paper outline.** The rest of this paper is structured as follows. Section 2 reviews some preliminaries in SDPs and PPM. Section 3 establishes quadratic growth and error bound properties. Section 4 establishes the linear convergence of primal and dual iterates of ALMs under standard assumptions. Section 5 presents numerical experiments, and Section 6 concludes the paper. While all results are presented for SDPs, analogous versions exist for other conic programs, such as LPs and SOCPs.

## 2 Preliminaries

### 2.1 Strong duality and strict complementarity

In this paper, we assume the linear mapping $\mathcal{A}$ to be *surjective* (or $\mathcal{A}^*$ is *injective*, thus $y$ and $Z$ have a unique correspondence). This is equivalent to $A_1, \cdots, A_m$ being linear independent. We here introduce two important regularity conditions: *strong duality* and *strict complementarity*.

**Definition 1** (Strong duality). We say (P) and (D) satisfies *strong duality* if we have $p^\star = d^\star$.

Strong duality only requires the optimal values to be the same. The existence of optimal primal and dual variables $(X^\star, y^\star, Z^\star)$ is ensured by the primal and dual *Slater's* condition [8, Theorem 3.1], i.e., there exist primal and dual feasible points $X$ and $y$ satisfying $X \in \text{int}(\mathbb{S}_+^n)$ and $C - \mathcal{A}^*(y) \in \text{int}(\mathbb{S}_+^n)$. Let $\Omega_P \subseteq \mathbb{S}^n$ and $\Omega_D \subseteq \mathbb{R}^m \times \mathbb{S}^n$ denote the optimal solutions of (P) and (D), respectively,

$$\Omega_P = \{X \in \mathbb{S}^n \mid \langle C, X \rangle = p^\star, \, \mathcal{A}(X) = b, X \in \mathbb{S}_+^n\}, \tag{1a}$$

$$\Omega_D = \{(y, Z) \in \mathbb{R}^m \times \mathbb{S}^n \mid \langle b, y \rangle = d^\star, \, \mathcal{A}^*(y) + Z = C, Z \in \mathbb{S}_+^n\}. \tag{1b}$$

Under the primal and dual *Slater's* condition, the solution sets $\Omega_P, \Omega_D$ are nonempty and compact (see e.g., [30, Section 2] or [29, Prop. 2.1]). We further have $\langle Z^\star, X^\star \rangle = 0, \forall X^\star \in \Omega_P, (y^\star, Z^\star) \in \Omega_D$, which is called *complementary slackness*.

Next, we introduce another regularity condition: *dual strict complementarity*, which is the key to ensuring many nice properties in a range of conic problems [34, 28, 35].

**Definition 2** (Dual strict complementarity for SDPs). Given a pair of optimal solutions $X^\star \in \Omega_P, (y^\star, Z^\star) \in \Omega_D$, we say that $(X^\star, y^\star, Z^\star)$ satisfies *dual strict complementarity* if $X^\star \in \text{relint}(\mathcal{N}_{(\mathbb{S}_+^n)^\circ}(-Z^\star))$, where $(\mathbb{S}_+^n)^\circ$ is the polar cone of $\mathbb{S}_+^n$, $\mathcal{N}_{(\mathbb{S}_+^n)^\circ}(-Z^\star)$ denotes the normal cone of $(\mathbb{S}_+^n)^\circ$ at $-Z^\star$, and $\text{relint}$ denotes relative interior. If (P) and (D) have one such pair, we say that (P) and (D) satisfy dual strict complementarity.

We note that the usual notion of *strict complementarity* for SDPs is equivalent to Definition 2, as pointed out in [28, Appendix B]. For completeness, we also review the standard notion of strict complementarity in Appendix A.1. The notion of (dual) strict complementarity is not restrictive, and it holds generically for conic programs [39], [40, Theorem 15]. Recently, it has been revealed that many structured SDPs from practical applications also satisfy strict complementarity [41]. One can also define *primal* strict complementarity, and the primal and dual strict complementarity are not equivalent in general, but they are equivalent for the so-called self-dual cones, such as $\mathbb{S}_+^n$.

## 2.2 Proximal point methods (PPMs)

It is now well-known that running ALMs for the primal (P) is equivalent to executing a PPM for the dual (D), and vice versa [23]. We here give a quick review of PPMs and their standard convergences.

Let $f : \mathbb{R}^n \to \bar{\mathbb{R}}$ be a proper closed convex function, and consider the problem $f^\star = \min_{x \in \mathbb{R}^n} f(x)$. Let $S = \text{argmin}_x \ f(x)$, which we assume to be nonempty. We define the *proximal operator* as

$$\text{prox}_{\alpha, f}(x) := \underset{y \in \mathbb{R}^n}{\text{argmin}} \ f(y) + \frac{1}{2\alpha} \|y - x\|^2, \tag{2}$$

where $\alpha > 0$. Starting with an initial point $x_0 \in \mathbb{R}^n$, the PPM generates a sequence of points as follows $x_{k+1} = \text{prox}_{c_k, f}(x_k)$, $k = 0, 1, 2, \ldots$, where $\{c_k\}_{k \geq 0}$ is a sequence of positive numbers bounded away from zero. The proximal operator (2) is always well-defined thanks to strong convexity.

In practice, the proximal operator (2) may not be easily evaluated, and thus an inexact version is often used [42]. In particular, an inexact PPM generates a sequence of points as

$$x_{k+1} \approx \text{prox}_{c_k, f}(x_k), \quad k = 0, 1, 2, \ldots. \tag{3}$$

Two classical types of inexactness, originating from the seminal work [42], are

$$\|x_{k+1} - \text{prox}_{c_k, f}(x_k)\| \leq \epsilon_k, \quad \sum_{k=0}^{\infty} \epsilon_k < \infty, \tag{a}$$

$$\|x_{k+1} - \text{prox}_{c_k, f}(x_k)\| \leq \delta_k \|x_{k+1} - x_k\|, \quad \sum_{k=0}^{\infty} \delta_k < \infty, \tag{b}$$

where $\{\epsilon_k\}_{k \geq 0}$ and $\{\delta_k\}_{k \geq 0}$ are two sequences of inexactness for (3). The classical work [42] has established asymptotic convergence of the iterates $x_k$ from (3) when using criterion (a). Fast linear convergences have also been established under various regularity conditions, such as the inverse of the subdifferential $(\partial f)^{-1}$ is locally Lipschitz at 0 [42] (or relaxed to be upper Lipschitz continuous at 0 [25]), and more generally $\partial f$ is metrically subregular [27, 43] (or $f$ satisfies *quadratic growth* [34, 44]). Recall that $f$ satisfies *quadratic growth* at $x^\star \in S$ if there exists a constant $\mu_q > 0$ such that

$$f(x) - f^\star \geq \mu_q/2 \cdot \text{dist}^2(x, S), \ \forall x \in \mathcal{U}, \tag{QG}$$

where $\mathcal{U}$ is a neighborhood containing the optimal solution $x^\star$. Convergence results for (3) are:

**Lemma 1.** *Let $S = \text{argmin}_{x \in \mathbb{R}^n} \ f(x)$ and suppose $S \neq \emptyset$. Let $\{x_k\}_{k \geq 0}$ be a sequence from (3).*

*(a) If criterion (a) is used, then $\lim_{k \to \infty} x_k = x_\infty \in S$, i.e., $x_k$ converges to an optimal solution.*

*(b) In addition to (a), if (b) is used and $f$ satisfies (QG) at $x_\infty \in S$, then there exists $\hat{k} > 0$ such that $\text{dist}(x_{k+1}, S) \leq \hat{\theta}_k \text{dist}(x_k, S), \forall k \geq \hat{k}$, where $\hat{\theta}_k = \frac{\theta_k + 2\delta_k}{1 - \delta_k}$ with $\theta_k = 1/\sqrt{c_k \mu_q + 1} < 1$.*

The first asymptotic convergence is taken from [42, Theorem 1], and different proofs for the second linear convergence under (QG) are available, and a simple one can be found in [44, Theorem 5.3]. Criterion (a) guarantees the iterate $x_k$ will arrive within a neighbor $\mathcal{U}$ where (QG) holds after some number $\hat{k}$, then criterion (b) ensures the linear convergence; see [44, Theorem 5.3] for details.

Note that Lemma 1 only guarantees the (linear) convergence of iterates $x_k$, which corresponds to the dual iterates in the ALM [23] (see Lemma 4 in Section 4). Thus, once a suitable condition like (QG) is established for (P) and (D), applying the ALM will enjoy linear convergence for dual iterates [21, 25, 26]. In this paper, we aim to establish linear convergences of both primal and dual iterates in the ALM, when applied to (P) or (D). For this, we first establish quadratic growth in Section 3.

## 3 Quadratic Growth in SDPs

In this section, we identify favorable quadratic growth and error bound properties for both primal and dual SDPs in (P) and (D), under the usual assumptions of strong duality and strict complementarity (cf. Definitions 1 and 2). These favorable structural properties will allow us to establish linear convergences of both primal and dual iterates of inexact ALMs applied to either (P) or (D).

### 3.1 Growth and error bound properties

To analyze PPM or ALM, we consider an equivalent reformulation of (P) or (D) in the following form

$$f_{\mathrm{P}}^{\star} := \min_{X \in \mathbb{S}^n} \; f_{\mathrm{P}}(X) := \langle C, X \rangle + g(X) \qquad (4)$$
$$\text{subject to} \quad \mathcal{A}(X) = b,$$

$$f_{\mathrm{D}}^{\star} := \min_{y \in \mathbb{R}^m, Z \in \mathbb{S}^n} \; f_{\mathrm{D}}(y, Z) := \langle -b, y \rangle + h(Z) \qquad (5)$$
$$\text{subject to} \quad \mathcal{A}^*(y) + Z = C,$$

where $g$ (and $h$, respectively) is chosen as either an indicator function $\delta_{\mathbb{S}_+^n}$ or an exact penalty function, defined as[2]

$$g(X) = \rho \max\{0, \lambda_{\max}(-X)\} \qquad \text{with } \rho > \mathbf{tr}(Z^\star). \qquad (6)$$

where $(y^\star, Z^\star) \in \Omega_{\mathrm{D}}$ can be any optimal solution. In (5), the exact penalty function $h$ is designed in the same way as (6) where the lower bound on $\rho$ is replaced by $\rho > \mathbf{tr}(X^\star)$ and $X^\star$ is any optimal solution in $\Omega_{\mathrm{P}}$. The common feature in (4) and (5) is to move the difficult conic constraint $X \in \mathbb{S}_+^n$ and $Z \in \mathbb{S}_+^n$ as a nonsmooth term $g(X)$ and $h(Z)$ in the cost function. It is clear that for both indicator and exact penalty functions, we have $g(X) = 0, \forall X \in \mathbb{S}_+^n$ and $h(Z) = 0, \forall Z \in \mathbb{S}_+^n$. For clarity, we state the following technical result.

**Lemma 2.** *Suppose strong duality holds for* (P) *and* (D). *Under the choice of an indicator function or an exact penalty function in* (6), *the set of optimal solutions for* (4) *is the same as* $\Omega_{\mathrm{P}}$ *in* (1a), *and the set of optimal solutions for* (5) *is the same as* $\Omega_{\mathrm{D}}$ *in* (1b).

The equivalence with an indicator function is obvious, and the equivalence with an exact penalty function has appeared in [30] and [29, Propositions 3.1&3.2]. Their proofs rely on a general characterization in exact penalty methods [45, Theorem 7.21]. In Appendix B, we present a simple, different, and self-contained proof without relying on prior results. Our proof uses a new characterization of the subdifferential of exact penalty functions (Proposition B.1), which might be of independent interest.

Our first main theoretical results are the following growth and error bound properties for primal and conic programs. The proofs will be outlined in Section 3.2.

**Theorem 1** (Growth properties in the primal). *Suppose strong duality and dual strict complementarity hold for* (P) *and* (D). *Consider the primal conic programs* (P) *and* (4). *Let* $(\bar{X}^\star, \bar{y}^\star, \bar{Z}^\star)$ *satisfy strict complementarity, and the exact penalty parameter is chosen as* $\rho > \mathbf{tr}(\bar{Z}^\star)$. *For any compact set* $\mathcal{U} \subseteq \mathbb{S}^n$ *containing an optimal solution* $X^\star \in \Omega_{\mathrm{P}}$, *there exist positive constants* $\kappa, \gamma, \alpha > 0$ *such that*

$$f_{\mathrm{P}}(X) - p^\star + \gamma \|\mathcal{A}(X) - b\| \geq \kappa \cdot \mathrm{dist}^2(X, \Omega_{\mathrm{P}}), \quad \forall X \in \mathcal{U}, \qquad (7a)$$

$$\langle C, X \rangle - p^\star + \gamma \|\mathcal{A}(X) - b\| + \alpha \cdot \mathrm{dist}(X, \mathbb{S}_+^n) \geq \kappa \cdot \mathrm{dist}^2(X, \Omega_{\mathrm{P}}), \quad \forall X \in \mathcal{U}, \qquad (7b)$$

*If* $\Omega_{\mathrm{P}}$ *is further compact, then set* $\mathcal{U}$ *in* (7a) *and* (7b) *can be chosen as any sublevel set of* $f_{\mathrm{P}}$, *i.e.,* $\{X \in \mathbb{S}^n \mid f_{\mathrm{P}}(X) \leq \beta, \mathcal{A}(X) = b\}$ *with a finite* $\beta$.

In (7a), the impact of the conic constraint $X \in \mathbb{S}_+^n$ is reflected in the reformulated cost $f_{\mathrm{P}}(X)$. When using the indicator function as $f_{\mathrm{P}}(X) = \langle C, X \rangle + \delta_{\mathbb{S}_+^n}(X)$, the left-hand side of (7a) implicitly requires $X \in \mathbb{S}_+^n$; otherwise $f_{\mathrm{P}}(X) = +\infty$ and thus the inequality (7a) holds trivially. If using the exact penalty function (6), we have $f_{\mathrm{P}}(X) = \langle C, X \rangle + g(X) < +\infty$ for $X \notin \mathbb{S}_+^n$, which is closer to (7b). Indeed, our proof of (7b) utilizes a characterization of the exact penalty function (6).

Both (7a) and (7b) give useful bounds when $X$ moves away from the optimal solution set $\Omega_{\mathrm{P}}$ in terms of suboptimality $f_{\mathrm{P}}(X) - p^\star$ or $\langle C, X \rangle - p^\star$, affine feasibility $\|\mathcal{A}(X) - b\|$, and conic feasibility $\mathrm{dist}(X, \mathbb{S}_+^n)$. They are closely related to *quadratic growth* in (QG), and are also referred to *error bound* properties [46, 33, 28]. The error bound plays a vital role in proving fast linear convergence for many iterative algorithms [34, 35, 44, 47]. Indeed, some previous studies [27–30] have established similar results to Theorem 1 for conic optimization, especially SDPs. As we will detail in Remark 1 below, our results are more general and unified. Similarly, the growth and error bound properties also hold for the dual conic program.

**Theorem 2** (Growth properties in the dual). *Suppose strong duality and dual strict complementarity hold for* (P) *and* (D). *Consider the dual problems* (D) *and* (5). *Let* $(\bar{X}^\star, \bar{y}^\star, \bar{Z}^\star)$ *satisfy strict complementarity, and the penalty parameter is chosen as* $\rho > \mathbf{tr}(\bar{X}^\star)$. *For any compact set* $\mathcal{V} \subseteq \mathbb{R}^m \times \mathbb{S}^n$ *containing an optimal solution* $(y^\star, Z^\star) \in \Omega_{\mathrm{D}}$, *there exist constants* $\kappa, \gamma, \alpha > 0$ *such that*

$$f_{\mathrm{D}}(y, Z) - (-d^\star) + \gamma \|C - \mathcal{A}^*(y) - Z\| \geq \kappa \cdot \mathrm{dist}^2((y, Z), \Omega_{\mathrm{D}}), \quad \forall (y, Z) \in \mathcal{V}, \qquad (8a)$$

---

[2]This penalty parameter $\rho$ guarantees that (4) and (P) are equivalent. To further ensure the dual problem of (P) is equivalent to the dual of (4), we need to choose $\rho > \sup_{(y^\star, Z^\star) \in \Omega_{\mathrm{D}}} \mathbf{tr}(Z^\star)$. This subtle point is less emphasized in the literature [30, 29]; see Remark B.1 in the appendix for details.

$$d^\star - \langle b, y \rangle + \gamma \|C - \mathcal{A}^*(y) - Z\| + \alpha \cdot \mathrm{dist}(Z, \mathbb{S}^n_+) \geq \kappa \cdot \mathrm{dist}^2((y, Z), \Omega_\mathrm{D}), \quad \forall (y, Z) \in \mathcal{V}. \quad \text{(8b)}$$

*If $\Omega_\mathrm{D}$ is further compact, then set $\mathcal{V}$ can be chosen as any sublevel set of $f_\mathrm{D}$, i.e., $\{(y, Z) \in \mathbb{R}^m \times \mathbb{S}^n \mid f_\mathrm{D}(y, Z) \leq \beta, C + Z = \mathcal{A}^*(y)\}$ with a finite $\beta$.*

The compactness of $\Omega_\mathrm{P}$ and $\Omega_\mathrm{D}$ in Theorems 1 and 2 is not restrictive, as it can be ensured by the dual and primal Slater's conditions (and $\mathcal{A}^*$ is injective) respectively; see e.g., [29, Proposition 2.1].

*Remark* 1. Our results in Theorems 1 and 2 are more general and unified than previous ones [27–30]. Theorems 1 and 2 hold for both the indicator function and the penalty function (6), while all previous results [27–30] have only discussed either one of them. We provide more comparisons below. The guarantees in (7a) and (8b) directly lead to a *quadratic growth* property of $f_\mathrm{P}$ and $f_\mathrm{D}$:

$$f_\mathrm{P}(X) - p^\star \geq \kappa \cdot \mathrm{dist}^2(X, \Omega_\mathrm{P}), \; \forall X \in \mathcal{U} \cap \{X \in \mathbb{S}^n \mid \mathcal{A}(X) = b\}, \quad \text{(9a)}$$

$$f_\mathrm{D}(y, Z) + d^\star \geq \kappa \cdot \mathrm{dist}^2((y, Z), \Omega_\mathrm{D}), \forall (y, Z) \in \mathcal{V} \cap \{(y, Z) \in \mathbb{R}^m \times \mathbb{S}^n \mid \mathcal{A}^*(y) + Z = C\}. \quad \text{(9b)}$$

When considering indicator functions, this property (9) is more general than [27, Corollary 3.1] since we allow for any compact set $\mathcal{U}$ (and also any sublevel set of $f_\mathrm{P}$ or $f_\mathrm{D}$). In contrast, the size of the neighborhood around an optimal solution $X^\star$ in [27, Corollary 3.1] depends on the eigenvalues of an optimal dual solution $Z^\star$. Thus the neighborhood only exists but its size might be unknown. Furthermore, our results (7a) and (8a) allow for the residual of the affine constraints $\mathcal{A}(X) = b$ and $C - \mathcal{A}^*(y) = Z$, while [27, Corollary 3.1] requires strict affine feasible points.

When considering exact penalty functions, similar quadratic growth properties appear in [29, 30]. However, the penalty $\rho$ in [29, 30] needs to be as large as $\rho \geq 2 \sup_{(y^\star, Z^\star) \in \Omega_\mathrm{D}} \mathbf{tr}(Z^\star)$. Our results in Theorems 1 and 2 only require a smaller constant as $\rho > \mathbf{tr}(\bar{Z}^\star)$ where $\bar{Z}^\star$ is an optimal dual solution satisfying strict complementarity, and this result is established via a new and simple argument (see Proposition B.2 in the appendix). Finally, an error bound similar to (7b) was also established in [28, Corollary 1] under similar assumptions of strong duality and strict complementarity. Our error bound (7b) does not require the absolute value of $f_\mathrm{P}(X) - p^\star$, and our proof strategy relies on a new characterization of exact penalty functions (see Lemma 3), which offers a new (possibly simpler) perspective than [28, Corollary 1]. The dual property (8b) is new and not discussed in [28]. $\quad\square$

*Remark* 2 (Conic programs). Theorems 1 and 2 are stated specifically for SDPs (P) and (D). It is known that linear programs (LP) and second-order cone programs (SOCP) are special cases of SDPs. Therefore, results similar to Theorems 1 and 2 when replacing $\mathbb{S}^n_+$ by nonnegative orthant or second-order cone also exist. Theorems 1 and 2 hold for a general class of conic programs. $\quad\square$

### 3.2 Proof sketches for Theorems 1 and 2

We here provide key proof sketches for Theorems 1 and 2. One key step is the following growth properties of an indicator function and a penalty function.

**Lemma 3.** *Let $\bar{X}, \bar{Z} \in \mathbb{S}^n_+$ satisfying $\langle \bar{X}, \bar{Z} \rangle = 0$ (i.e., both $\bar{X}$ and $\bar{Z}$ are on the boundary unless one of them is zero). If $l : \mathbb{S}^n \to \bar{\mathbb{R}}$ is either an indicator function $l = \delta_{\mathbb{S}^n_+}$, or a penalty function*

$$l(X) = \rho \max\{0, \lambda_{\max}(-X)\} \; \text{ with } \rho > \mathbf{tr}(\bar{Z}), \quad \text{(10)}$$

*then we have that*

$$(\partial l)^{-1}(-\bar{Z}) = (\partial \delta_{\mathbb{S}^n_+})^{-1}(-\bar{Z}) = \mathcal{N}_{(\mathbb{S}^n_+)^\circ}(-\bar{Z}). \quad \text{(11)}$$

*Moreover, for any positive value $\mu \in (0, \infty)$, there exists a positive constant $\kappa$ such that*

$$l(X) \geq l(\bar{X}) + \langle -\bar{Z}, X - \bar{X} \rangle + \kappa \cdot \mathrm{dist}^2\left(X, (\partial l)^{-1}(-\bar{Z})\right), \quad \forall X \in \mathbb{B}(\bar{X}, \mu). \quad \text{(12)}$$

Note that $l$ in (10) is related to, but not identical to, the exact penalty function in (6). The proof of Lemma 3 heavily exploits the nice self-dual structure of the cone $\mathbb{S}^n_+$. For better clarity, we provide the proofs of (11) and (12) in Proposition B.1 and Appendix C respectively.

*Remark* 3. We believe that the results in Lemma 3 are of independent interest, especially in terms of the penalty function (10). Building upon (12), one can further show $l$ is *metrically subregular* [48, Theorem 3.3]. The most closely related characterization is [27, Proposition 3.3], which focuses solely on the indicator function of the positive semidefinite cone $\mathbb{S}^n_+$. However, even in this scenario, the result in [27, Proposition 3.3] only guarantees the existence of a small neighborhood around $\bar{X}$. In contrast, our result (12) works for any closed ball $\mathbb{B}(\bar{X}, \mu)$ around $\bar{X}$ with a finite radius $\mu > 0$. $\quad\square$

Upon establishing Lemma 3, with bounded linear regularity (see Appendix A.4) that is guaranteed by dual strict complementarity, we can establish the following growth properties for (4) and (5).

**Proposition 1.** *Consider primal and dual conic programs* (P) *and* (D) *and their equivalent forms in* (4) *and* (5) *respectively. Suppose strong duality and dual strict complementarity hold for* (P) *and* (D). *Let* $(X^\star, y^\star, Z^\star)$ *be a pair of primal and dual optimal solutions, i.e.,* $X^\star \in \Omega_P$ *and* $(y^\star, Z^\star) \in \Omega_D$. *For any* $\mu > 0$, *there exist positive constants* $\kappa_1, \gamma_1, \kappa_2, \gamma_2$ *such that*

$$f_P(X) - f_P^\star + \gamma_1 \|\mathcal{A}(X) - b\| \geq \kappa_1 \cdot \mathrm{dist}^2(X, \Omega_P), \ \forall X \in \mathbb{B}(X^\star, \mu),$$

$$f_D(y, Z) - f_D^\star + \gamma_2 \|C - \mathcal{A}^*(y) - Z\| \geq \kappa_2 \cdot \mathrm{dist}^2((y, Z), \Omega_D), \ \forall (y, Z) \in \mathbb{B}((y^\star, Z^\star), \mu).$$

The proof of Proposition 1 is given in Appendix D.1. We now see that Theorems 1 and 2 are direct consequences of Proposition 1 as $\max\{0, \lambda_{\max}(-X)\} \leq \mathrm{dist}(X, \mathbb{S}_+^n)$ for all $X \in \mathbb{S}^n$. The compactness statement in Theorems 1 and 2 comes from the fact that the solution set $\Omega_P$ (resp. $\Omega_D$) is compact if and only if any sublevel set of $f_P$ (resp. $f_D$) is compact (see, e.g., [29, Lemma D.1]).

# 4 Augmented Lagrangian Methods (ALMs) for SDPs

In this section, we prove that both primal and dual iterates of ALMs, when applied to (P) and (D), enjoy linear convergence under the usual assumption of strict complementarity.

## 4.1 Inexact Augmented Lagrangian Method

In principle, ALM can be applied to solve both (P) and (D). However, most existing results focus on solving (D) [18, 19]. For simplicity, we here focus on one formulation of the augmented Lagrangian function for (P). The dual formulation is presented in Appendix E.3 for completeness.

We start by introducing two dual variables $y \in \mathbb{R}^m$ and $Z \in \mathbb{S}_+^n$ and defining the Lagrangian function for (P) as $L_0(X, y, Z) = \langle C, X \rangle + \langle y, b - \mathcal{A}(X) \rangle + \langle Z, X \rangle$. The corresponding Lagrangian dual function and dual problem read as

$$g_0(y, Z) = \inf_{X \in \mathbb{S}^n} L_0(X, y, Z) \quad \text{and} \quad \max_{y \in \mathbb{R}^m, Z \in \mathbb{S}_+^n} g_0(y, Z). \tag{13}$$

It can be verified the dual in (13) is the same as the dual conic program (D). Given a penalty parameter $r > 0$, the *Augmented Lagrangian* function of (P) corresponding to $L_0$ is defined as

$$L_r(X, y, Z) = \langle C, X \rangle + \frac{1}{2r}(\|y + r(b - \mathcal{A}(X))\|^2 + \|\Pi_{\mathbb{S}_+^n}(Z - rX)\|^2 - \|y\|^2 - \|Z\|^2), \tag{14}$$

where $\Pi_{\mathbb{S}_+^n}(\cdot)$ denotes the orthogonal projection onto the cone $\mathbb{S}_+^n$. Note that $L_r$ is continuously differentiable as $\|\Pi_{\mathbb{S}_+^n}(\cdot)\|^2$ is continuously differentiable [18, Section 1]. Given initial points $(y_0, Z_0) \in \mathbb{R}^m \times \mathbb{S}_+^n$ and a sequence of positive scalars $r_k \uparrow r_\infty < +\infty$, the *inexact ALM* generates a sequence of $\{X_k\}$ (primal variables) and $\{y_k, Z_k\}$ (dual variables) as

$$X_{k+1} \approx \min_{X \in \mathbb{S}^n} L_{r_k}(X, y_k, Z_k), \tag{15a}$$

$$y_{k+1} = y_k + r_k \nabla_y L_{r_k}(X_{k+1}, y_k, Z_k) = y_k + r_k(b - \mathcal{A}(X_{k+1})), \tag{15b}$$

$$Z_{k+1} = Z_k + r_k \nabla_Z L_{r_k}(X_{k+1}, y_k, Z_k) = \Pi_{\mathbb{S}_+^n}(Z_k - r_k X_{k+1}). \tag{15c}$$

For notational convenience, we write $w = (y, Z)$ and $w_k = (y_k, Z_k)$ and consider the following two inexactness criteria for solving (15a) (since it can be a challenge to solve (15a) exactly):

$$L_{r_k}(X_{k+1}, w_k) - \min_{X \in \mathbb{S}^n} L_{r_k}(X, w_k) \leq \epsilon_k^2/(2r_k), \quad \sum_{k=1}^\infty \epsilon_k < \infty, \tag{A'}$$

$$L_{r_k}(X_{k+1}, w_k) - \min_{X \in \mathbb{S}^n} L_{r_k}(X, w_k) \leq \delta_k^2 \|w_{k+1} - w_k\|^2/(2r_k), \quad \sum_{k=1}^\infty \delta_k < \infty. \tag{B'}$$

The two stopping criteria above are suggested by the seminal work of Rockafellar [23].

## 4.2 Linear convergences of primal and dual iterates in ALM

After [23], one classical method for analyzing the convergence of the inexact ALM (15) is to utilize the connection between ALM and PPM. We review the following important lemma.

**Lemma 4** ([23, Proposition 6]). *The dual iterates $w_{k+1} = (y_{k+1}, Z_{k+1})$ in (15b) and (15c) satisfy the following relationship*

$$\|w_{k+1} - \text{prox}_{r_k, -g_0}(w_k)\|^2/(2r_k) \leq L_{r_k}(X_{k+1}, w_k) - \inf_{X \in \mathbb{S}^n} L_{r_k}(X, w_k),$$

*where $g_0$ is the Lagrangian dual function in (13).*

If (15a) is updated exactly, Lemma 4 confirms that $w_{k+1} = \text{prox}_{r_k, -g_0}(w_k)$: the dual iterate of the ALM agree with the PPM iterate for the dual problem (13). If (15a) is updated inexactly, the iterate $w_{k+1}$ *may not* satisfy the affine constraint $C = Z_{k+1} + \mathcal{A}^*(y_{k+1})$. Moreover, viewing Lemma 4, the stopping criteria (A$'$) and (B$'$) naturally imply that the dual iterate $w_{k+1}$ satisfies

$$\|w_{k+1} - \text{prox}_{r_k, -g_0}(w_k)\| \leq \epsilon_k \text{ and } \|w_{k+1} - \text{prox}_{r_k, -g_0}(w_k)\| \leq \delta_k \|w_{k+1} - w_k\|.$$

Then, we can establish the convergence of the dual sequence $\{w_k\}$ in (15) from Lemma 1 via PPM. In particular, this observation was first discovered in [23] and later tailored in convex composite conic programmings in [26]. We summarize asymptotic convergences of (15) below.

**Proposition 2** (Asymptotic convergences). *Consider* (P) *and* (D). *Assume strong duality holds and $\Omega_D \neq \emptyset$. Let $\{X_k, w_k\}$ be a sequence from the ALM (15) under* (A$'$). *The following hold.*

(a) *The dual sequence $\{w_k\}$ is bounded. Further, $\lim_{k \to \infty} w_k = w_\infty \in \Omega_D$ (i.e., the whole sequence converges to a dual optimal solution).*

(b) *The primal feasibility and cost value gap satisfy*

$$\text{dist}(X_{k+1}, \mathbb{S}^n_+) \leq \|Z_k - Z_{k+1}\|/r_k \to 0, \quad \|\mathcal{A}(X_{k+1}) - b\| = \|y_k - y_{k+1}\|/r_k \to 0,$$

$$\langle C, X_{k+1} \rangle - p^\star \leq L_{r_k}(X_{k+1}, w_k) - \min_{X \in \mathbb{S}^n} L_{r_k}(X, w_k) + (\|w_k\|^2 - \|w_{k+1}\|^2)/(2r_k) \to 0.$$

(c) *If the primal solution set $\Omega_P$ in (1a) is nonempty and bounded, then the primal sequence $\{X_k\}$ is also bounded, and all of its cluster points belongs to $\Omega_P$.*

*Proof.* We give the sketch of proof for parts (a) and (b). A complete proof can be found in Appendix E.1. Part (a) comes directly from the PPM convergence Lemma 1 as Lemma 4 and the stopping criteria (A$'$) naturally imply the dual iterate $w_{k+1}$ satisfies $\|w_{k+1} - \text{prox}_{r_k, -g_0}(w_k)\| \leq \epsilon_k$ and $\sum_{k=1}^{\infty} \epsilon_k < \infty$.

In part (b), the first inequality uses the fact that $Z_{k+1} + (X_{k+1} - X_k)/r_k \in \mathbb{S}^n_+$ by performing Moreau decomposition [49, Exercise 12.22] on $r_k X_{k+1} - Z_k$ and using the update rule (15b); The second inequality comes directly from (15c); The last inequality uses the definition of $L_{r_k}(X_{k+1}, w_k)$ in (14) and the fact $\min_{X \in \mathbb{S}^n} L_{r_k}(X, w_k) \leq p^\star$.

Part (c) is a consequence of part (b) and the fact that $\Omega_P$ is bounded if and only if for any $\gamma \in \mathbb{R}^3$ the set $\{X \in \mathbb{S}^n \mid \text{dist}(X, \mathbb{S}^n_+) \leq \gamma_1, \|\mathcal{A}(X) - b\| \leq \gamma_2, \langle C, X \rangle \leq \gamma_3\}$ is bounded [23, page 110]. $\square$

Proposition 2 establishes the asymptotic convergences for both the primal and dual variables. The linear convergence of the dual iterates can also be deduced when the negative of the dual $g_0$ defined in (13) satisfies (QG). However, the rate of the primal iterates remains unclear. Here, leveraging the error bound (8b), we also derive a linear convergence of the primal iterates, which is one main technical contribution of this work. We summarize linear convergence results below.

**Theorem 3** (Linear convergences). *Consider* (P) *and* (D). *Assume strong duality and dual strict complementarity holds (implying $\Omega_P \neq \emptyset$ and $\Omega_D \neq \emptyset$). Let $\{X_k, w_k\}$ be a sequence from the ALM (15) under* (A$'$) *and* (B$'$). *The following statements hold.*

(a) *(Dual iterates and KKT residuals) There exist constants $\hat{k} > 0$ such that for all $k \geq \hat{k}$, we have*

$$\text{dist}(w_{k+1}, \Omega_D) \leq \mu_k \cdot \text{dist}(w_k, \Omega_D), \quad \text{dist}(X_{k+1}, \mathbb{S}^n_+) \leq \mu'_k \cdot \text{dist}(w_k, \Omega_D),$$

$$\|\mathcal{A}(X_{k+1}) - b\| \leq \mu'_k \cdot \text{dist}(w_k, \Omega_D), \quad \langle C, X_{k+1} \rangle - p^\star \leq \mu''_k \cdot \text{dist}(w_k, \Omega_D), \tag{16}$$

*where $0 < \mu_k < 1$, $\mu'_k > 0$, and $\mu''_k > 0$ are positive finite constants.*

(b) *(Primal iterates) If $\Omega_P$ is bounded, then the primal sequence $\{X_k\}$ also converges linearly to the solution set $\Omega_P$, i.e., there a constant $\hat{k} > 0$ such that for all $k \geq \hat{k}$,*

$$\text{dist}^2(X_{k+1}, \Omega_P) \leq \tau_k \cdot \text{dist}(w_k, \Omega_D), \tag{17}$$

*where $\tau_k > 0$ is a finite constant.*

*Proof.* The proof of part (a) is largely motivated by the techniques in [26] that focus on ALMs on dual SDPs. Part (b) is a consequence of the error bound (7b). Here, we highlight the key steps of the proof of part (a) due to the page limit, and detailed arguments about part (a) can be found in Appendix E.2. Note that if dual strict complementarity holds, Theorem 2 (with $h$ in (5) as an indicator function) guarantees the following two nice properties: 1) $-g_0$ in (13) satisfies quadratic growth in (9b); 2) the error bound (7b) holds.

1. By the discussion after Lemma 4, since (A) and (B$'$) are in force, the dual sequence $\{w_k\}$ satisfies
$$\|w_{k+1} - \mathrm{prox}_{r_k,-g_0}(w_k)\| \leq \epsilon_k, \|w_{k+1} - \mathrm{prox}_{r_k,-g_0}(w_k)\| \leq \delta_k \|w_{k+1} - w_k\|, \forall k \geq 0.$$
Applying the convergence result of PPM in Lemma 1 yields the linear convergence of the dual distance, i.e., there exists a $k_1 \geq 0$ such that
$$\mathrm{dist}(w_{k+1}, \Omega_D) \leq \mu_k \cdot \mathrm{dist}(w_k, \Omega_D), \ \mu_k < 1, \ \forall k \geq k_1.$$
Using part (b) in Proposition 2 and $\|w_{k+1} - w_k\| \leq \frac{1}{1-\delta_k} \mathrm{dist}(w_k, \Omega_D)$ if $\delta_k < 1$, we can show
$$\max\{\mathrm{dist}(X_{k+1}, \mathbb{S}_+^n), \|\mathcal{A}(x_{k+1}) - b\|\} \leq \frac{1}{(1-\delta_k)r_k} \mathrm{dist}(w_k, \Omega_D),$$
$$\langle C, X_{k+1}\rangle - p^\star \leq \frac{\delta_k^2 \|w_{k+1} - w_k\| + \|w_k\| + \|w_{k+1}\|}{2r_k(1-\delta_k)} \mathrm{dist}(w_k, \Omega_D).$$
Finally, as $\delta_k \to 0$, there exists $k_2 \geq 0$ such that $\delta_k < 1$ and for all $k \geq k_2$. Thus, part (a) then follows from taking $\hat{k} = \max\{k_1, k_2\}$.

2. Note that the error bound (7b) and the assumption that $\Omega_P$ is bounded guarantee that for any bounded set $\mathcal{U}$ containing $\Omega_P$, there exist constants $\alpha_1, \alpha_2, \alpha_3 > 0$ such that
$$\langle C, X\rangle - \langle C, X^\star\rangle + \alpha_1 \|\mathcal{A}(X) - b\| + \alpha_2 \cdot \mathrm{dist}(X, \mathbb{S}_+^n) \geq \alpha_3 \cdot \mathrm{dist}^2(X, \Omega_P), \quad \forall X \in \mathcal{U}. \quad (18)$$
By Proposition 2 - part (c), the sequence $\{X_k\}$ is bounded. Let $\mathcal{U} \subseteq \mathbb{S}^n$ be a bounded set that contains $\Omega_P$ and all the iterates $\{X_k\}$. Combing (18) with (16), we have
$$\alpha_3 \cdot \mathrm{dist}^2(X_{k+1}, \Omega_P) \leq (\mu_k'' + (\alpha_1 + \alpha_2)\mu_k') \cdot \mathrm{dist}(w_k, \Omega_D), \quad \forall k \geq \hat{k}.$$
Dividing both sides by $\alpha_3$ leads to the desired result in part (b). This completes the proof. $\quad\square$

To our best knowledge, achieving linear convergence for the primal iterates of inexact ALMs requires a significantly stronger condition on the Lagrangian function [23, Theorem 5] and implementing an additional stopping criterion [26, Proposition 3]. Unfortunately, as highlighted in [26, Section 3], such an assumption in [23, Theorem 5] can easily fail for general conic programs.

Our result Theorem 3, however, reveals that the linear convergence of the primal iterates also happens under the standard assumption of strict complementarity and bounded primal solution set. This suggests that primal linear convergence is often expected since strict complementary is a generic property of conic programs [40, Theorem 15]. Our result not only completes theoretical convergences of inexact ALMs but also offers insights for practical successes in [18, 19].

*Remark* 4 (Strict complementarity). As we see in the proof of Theorem 3, strict complementarity is the key to ensure that 1) the negative of the function $g_0$ in (13) satisfies quadratic growth; 2) the error bound property (E.2) holds for the primal conic program. The quadratic growth is used to derive the dual linear convergence (16), and the error bound is to conclude the primal linear convergence. $\quad\square$

## 5 Numerical experiments

In this section, we present numerical experiments to examine the empirical performance of ALM introduced in Section 4. We consider two applications in combinational problems and machine learning, including the SDP relaxation of maximum cut (Max-Cut) problem [50] and linear Support Vector Machine (SVM). The Max-Cut problem and the linear SVM can be formulated as

$$\min_{X \in \mathbb{S}^n} \quad \langle C, X\rangle \qquad\qquad \min_{x \in \mathbb{R}^n, t \in \mathbb{R}^m} \quad \lambda \mathbf{1}^\mathsf{T} t + \frac{1}{2}\|x\|^2,$$
$$\text{subject to} \quad \mathrm{Diag}(X) = \mathbf{1}, \qquad\qquad \text{subject to} \quad \mathrm{diag}(b)Ax + \mathbf{1} \leq t,$$
$$X \in \mathbb{S}_+^n, \qquad\qquad\qquad\qquad t \in \mathbb{R}_+^m.$$

where $\mathbf{1}$ is an all one vector, $A \in \mathbb{R}^{m \times n}$ and $b \in \mathbb{R}^m$ are problem data in the linear SVM, $\mathrm{diag}(b)$

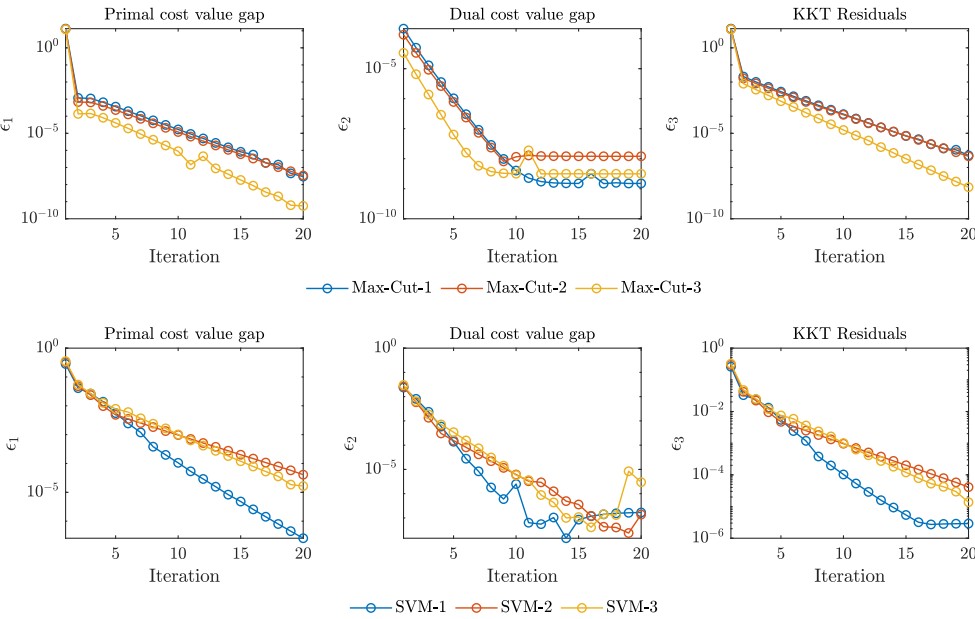

**Figure 1:** Numerical convergence behavior of inexact ALM (15) for Max-Cut and linear SVM. The symbol $\epsilon_3$ denotes the KKT residuals $\epsilon_3 = \max\{\eta_1, \eta_2, \eta_3, \eta_4, \eta_5\}$.

denotes the diagonal matrix with $b$ as the diagonal elements, and $\lambda > 0$ is a constant. We can see that Max-Cut is of the same form as (P). Despite linear SVM is not in the form of SDP (P) but a quadratic program, the corresponding ALM with similar convergence guarantees can also be derived since a quadratic program is a special case of SDP. For Max-Cut problem, we select the graph $G_1, G_2$, and $G_3$ from the website `https://web.stanford.edu/~yyye/yyye/Gset/` and only take the first $20 \times 20$ submatrix as the considered problem data $C$. For linear SVM, we randomly generate the problem data $A \in \mathbb{R}^{10 \times 100}$ and $b \in \{-1, 1\}^{100}$. We then apply the ALM (15) for those instances and compute the relative primal and dual cost value gap and the relative feasibility residuals as follows

$$\epsilon_1 = \frac{|\langle C, X_k \rangle - p^\star|}{1 + |p^\star|}, \epsilon_2 = \frac{|\langle b, y_k \rangle - d^\star|}{1 + |d^\star|}, \eta_1 = \frac{\|\mathcal{A}(X_k) - b\|}{1 + \|b\|}, \eta_2 = \frac{\|X_k - \Pi_{\mathbb{S}_+^n}(X_k)\|}{1 + \|X_k\|},$$

$$\eta_3 = \frac{\|C - \mathcal{A}^*(y_k) - Z_k\|}{1 + \|C\|}, \eta_4 = \frac{\|Z_k - \Pi_{\mathbb{S}_+^n}(Z_k)\|}{1 + \|Z_k\|}, \eta_5 = \frac{|\langle C, X_k \rangle - \langle b, y_k \rangle|}{1 + |d^\star|}.$$

The numerical results are presented in Figure 1. In all cases, the primal and dual cost value gap and the KKT residuals all converge linearly to at least the accuracy of $10^{-5}$. The oscillation or flattening behavior that appears in the tail (when the iterates are close to the solution set) could be due to the inaccuracy of the subproblem solver and computational impreciseness. A detailed theoretical analysis of such behavior is interesting, and we leave it to future work.

Further details on the algorithm parameters, problem instances, and more numerical experiments for machine learning applications can be found in Appendix F.

## 6 Conclusion

In this paper, we have established the quadratic growth and error bound of two different variants (4) and (5) of SDPs under the condition of strict complementarity. Central to our approach is the examination of the growth properties of the indicator and exact penalty functions. By leveraging these new quadratic growth and error bounds, we establish the linear convergence of both primal and dual iterates of inexact ALMs when applied to semidefinite programs, under the usual assumption of strict complementarity and a bounded solution set on the primal side. Our result not only fills a void in the convergence theory of inexact ALMs but also offers valuable insights into the exceptional numerical performance of solvers rooted in ALMs, such as SDPNAL+ [19]. We expect further interesting applications in machine learning and polynomial optimization [11] for inexact ALMs.

## Acknowledgments and Disclosure of Funding

This work is supported by NSF ECCS 2154650, NSF CMMI 2320697, and NSF CAREER 2340713. We would like to thank Dr. Xin Jiang for the discussions on exact penalty functions, and Mr. Pranav Reddy for his assistance in proofreading the manuscript and contributions to the appendix.

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

# Contents

# Appendix

The appendix is divided into the following seven sections:

1. Appendix A presents some preliminaries in conic optimization and standard ALMs.
2. Appendix B presents a self-contained proof for the exact penalty functions and a new characterized of the preimage of the exact penalty functions.
3. Appendix C completes the proof of growth properties of the indicator function and exact penalty function.
4. Appendix D details the proof of Proposition 1 and presents a simple example to illustrate the growth property.
5. Appendix E.2 completes the missing proofs in Theorem 3.
6. Appendix E.3 discusses the convergence of ALMs applied to the dual problem (D).
7. Appendix F presents further details of numerical experiments in Section 5.

## A  Optimization Background

**Notations**  For clarity, we summarize common notations here. We use $\mathbb{R}^n$ to denote the $n$-dimensional real space and $\mathbb{S}^n$ to denote the space of real $n \times n$ symmetric matrices. The cone of nonnegative orthant is denoted as $\mathbb{R}^n_+ = \{x \in \mathbb{R}^n \mid x_i \geq 0, \forall i = 1, \ldots, n\}$. The cone of positive and negative semidefinite matrices are defined as $\mathbb{S}^n_+ = \{X \in \mathbb{S}^n \mid v^\mathsf{T} X v \geq 0, \forall v \in \mathbb{R}^n\}$ and $\mathbb{S}^n_- = -\mathbb{S}^n_+$ respectively. For vectors in $\mathbb{R}^n$, the inner product $\langle \cdot, \cdot \rangle$ is defined as $\langle x, y \rangle = x^\mathsf{T} y$ and the normal $\| \cdot \|$ stands for the $l_2$ norm. For matrices in $\mathbb{S}^n$, the inner product $\langle \cdot, \cdot \rangle$ is defined as $\langle X, Y \rangle = \mathbf{tr}(X^\mathsf{T} Y)$ and the norm $\| \cdot \|_{\mathrm{op}}, \| \cdot \|_*$ and $\| \cdot \|$ denote the operator two norm, nuclear norm, and Frobenious norm respectively. Given a set $S \subseteq \mathbb{R}^n$, the relative interior of $S$ is denoted as $\mathrm{relint}(S)$, and $\delta_S(x)$ is defined as the indicator function of $S$, i.e., $\delta_S(x) = 0$ if $x \in S$ and $\delta_S(x) = \infty$ otherwise. The closed ball with a radius $\mu$ around the center $x \in \mathbb{R}^n$ is written as $\mathbb{B}(x, \mu) = \{y \in \mathbb{R}^n \mid \|y - x\| \leq \mu\}$.

### A.1  Standard notion of strict complementarity in SDPs

In this subsection, we review the standard algebraic notion definition for SDPs which is equivalent to Definition 2. A proof of the equivalence can be found in [28, Appendix B].

**Definition A.1** (Strict complementarity for SDPs). Consider the pair of primal and dual SDPs (P) and (D). We say the strict complementarity holds if there exists a pair of primal and dual solutions $(X^\star, y^\star, Z^\star)$ satisfying

$$X^\star + Z^\star \in \mathbb{S}^n_{++} \text{ or } \mathrm{rank}(X^\star) + \mathrm{rank}(Z^\star) = n. \tag{A.1}$$

With the complementarity slackness $\langle X^\star, Z^\star \rangle = 0$, we have $X^\star Z^\star = Z^\star X^\star = 0$. This means there exists an orthonormal matrix $Q \in \mathbb{R}^{n \times n}$ with $Q^\mathsf{T} Q = I$, such that

$$X^\star = Q \cdot \mathrm{diag}(\lambda_1, \ldots, \lambda_n) \cdot Q^\mathsf{T}, \quad Z^\star = Q \cdot \mathrm{diag}(w_1, \ldots, w_n) \cdot Q^\mathsf{T} \tag{A.2}$$

and $\lambda_i w_i = 0, i = 1, \ldots, n$. Then, (A.1) implies that exactly one of the two conditions $\lambda_i = 0$ and $w_i = 0$ is true in (A.2). In other words, the eigenvalues of $X^\star$ and $Z^\star$ have a complementary sparsity pattern.

*Remark* A.1 (Dual strict complementarity in the general case). The dual strict complementarity in Definition 2 is consistent with [28, Defnition 2]. A variant of dual strict complementarity in the general case is given in [34, Section 4], which requires that

$$0 \in \mathrm{relint}(\partial g(y^\star)), \tag{A.3}$$

where $g(y) = \langle b, y \rangle - \delta_{\mathbb{S}^n_+}(C - \mathcal{A}^*(y))$. Definition 2 is also consistent with (A.3). Indeed, let a dual optimal solution $(y^\star, Z^\star)$ satisfy (A.3). Considering $\partial g(y) = b + \mathcal{A}(\mathcal{N}_{\mathbb{S}^n_+}(C - \mathcal{A}^*(y)))$ and using [51, Corollary 6.6.2 and Theorem 6.6], we have

$$0 \in \mathrm{relint}(b + \mathcal{A}(\mathcal{N}_{\mathbb{S}^n_+}(Z^\star))) = b + \mathrm{relint}(\mathcal{A}(\mathcal{N}_{\mathbb{S}^n_+}(Z^\star))) = b + \mathcal{A}(\mathrm{relint}(\mathcal{N}_{\mathbb{S}^n_+}(Z^\star))),$$

which means there exists a point $s \in \mathrm{relint}(\mathcal{N}_{\mathbb{S}^n_+}(Z^\star))$ such that $0 = b + \mathcal{A}(s)$. Equivalently, there exists $X^\star = -s \in \mathrm{relint}(\mathcal{N}_{(\mathbb{S}^n_+)^\circ}(-Z^\star))$ (recall that $(\mathbb{S}^n_+)^\circ = -\mathbb{S}^n_+$) such that $\mathcal{A}(X^\star) = b$. This is equivalent to Definition 2. $\qquad\square$

## A.2 Augmented Lagrangian Methods (ALMs) for inequality constraints

We here review some classic results of ALM. Consider a constrained convex optimization problem

$$\min_{x \in \mathcal{X}_0} \quad f(x)$$
$$\text{subject to} \quad f_i(x) \leq 0, \quad \forall i = 1, 2, \ldots, m, \tag{A.4}$$

where $\mathcal{X}_0 \subseteq \mathbb{R}^n$ is a closed convex set and $f, f_i : \mathbb{E} \to \bar{\mathbb{R}}$ are proper closed convex functions. We associate each inequality constraint in (A.4) with a dual variable $z_i \geq 0$, and the Lagrangian function for (A.4) is $L_0(x, z) = f(x) + \sum_{i=1}^m z_i f_i(x)$. The corresponding dual function is

$$g_0(z) = \inf_{x \in \mathcal{X}_0} L_0(x, z). \tag{A.5}$$

One way to define the augmented Lagrangian for (A.4) is to transform the inequality constraints into equality constraints [45, Chapter 4.7]. After some manipulation (see [45, Equation 4.79] or Appendix A.3), the classical augmented Lagrangian with parameter $r > 0$ for (A.4) reads as

$$L_r(x, z) = f(x) + \frac{1}{2r} \left( \|\Pi_{\mathbb{R}_+^m}(z + rf(x))\|^2 - \|z\|_2^2 \right), \tag{A.6}$$

where $\Pi_{\mathbb{R}_+^m}(\cdot)$ denotes the Euclidean projection onto $\mathbb{R}_+^m$. Note that $L_r$ is differentiable in $z$ [52]. Given an initial point $z_0 \in \mathbb{R}^m$ and a sequence of positive scalars $r_0 \uparrow r_\infty < +\infty$, the inexact ALM generates a sequence of $\{x_k\}$ (primal variables) and $\{z_k\}$ (dual variables) as

$$x_{k+1} \approx \min_{x \in \mathcal{X}_0} L_{r_k}(x, z_k), \tag{A.7a}$$

$$z_{k+1} = z_k + r_k \nabla_z L_{r_k}(x_{k+1}, z_k) = \Pi_{\mathbb{R}_+^m}(z_k + r_k f(x_{k+1})), \quad k = 0, 1, \ldots. \tag{A.7b}$$

The convergence of $x_k$ and $z_k$ depends on the inexactness in (A.7a). For this, the seminal work by Rockafellar [23] points out an important connection between PPM and ALM, summarized below.

**Lemma A.1** ([23, Proposition 6])**.** *The dual iterate $z_{k+1}$ in* (A.7b) *satisfies the following relationship*

$$\|z_{k+1} - \text{prox}_{r_k, -g_0}(z_k)\|^2 / (2r_k) \leq \bar{L}_{r_k}(x_{k+1}, z_k) - \inf_{x \in \mathcal{X}_0} L_{r_k}(x, z_k),$$

*where $g_0$ is the Lagrangian dual function in* (A.5)*, and $\bar{L}_r$ denotes an extended form of the augmented Lagrangian $L_r$ as $\bar{L}_r(x, z) = L_r(x, z)$, if $x \in \mathcal{X}_0$, otherwise $\bar{L}_r(x, z) = \infty$.*

Note the subtle difference between Lemmas 4 and A.1. If (A.7a) is updated exactly, Lemma A.1 confirms that $z_{k+1} = \text{prox}_{c_k, -g_0}(z_k)$. In other words, the dual iterates of the ALM are the same as the iterates from PPM on the dual problem (A.5). Moreover, by controlling the inexactness (A.7a), the convergence of the dual variables $z_k$ can be naturally guaranteed from Lemma 1. Motivated by (a) and (b), the following two criteria are commonly used in (A.7a)

$$\bar{L}_{c_k}(x_{k+1}, z_k) - \min_{x \in \mathcal{X}_0} \bar{L}_{r_k}(x, z_k) \leq \epsilon_k^2 / (2r_k), \quad \textstyle\sum_{k=1}^\infty \epsilon_k < \infty, \tag{A}$$

$$\bar{L}_{c_k}(x_{k+1}, z_k) - \min_{x \in \mathcal{X}_0} \bar{L}_{r_k}(x, z_k) \leq \delta_k^2 \|z_{k+1} - z_k\|^2 / (2r_k), \quad \textstyle\sum_{k=1}^\infty \delta_k < \infty, \tag{B}$$

where $\{\epsilon_k\}_{k \geq 0}$ and $\{\delta_k\}_{k \geq 0}$ are two sequences of inexactness. Note that (A) and (B) guarantee the dual variables $z_k$ from ALM (A.7) satisfy (a) and (b) with the inexact PPM (3), respectively. Let $S = \text{argmax}_{z \geq 0} g_0(z)$ denote the set of optimal dual solutions for (A.4). Combining Lemma A.1 with Lemma 1, the iterates from the inexact ALM (A.7) has the following convergence guarantees.

**Lemma A.2.** *Assume $S \neq \emptyset$. Let $\{x_k, z_k\}$ be a sequence generated by the inexact ALM* (A.7)*.*

*(a) If criterion* (A) *is used, then $\lim_{k \to \infty} z_k = z_\infty \in S$ (i.e., $z_k$ converges to a dual optimal solution), and the primal sequence $\{x_k\}$ satisfies $f_i(x_{k+1}) \leq \|z_k - z_{k+1}\| / c_k, i = 1, \ldots, m$ (primal feasibility), and $\langle c, x_{k+1} \rangle - p^\star \leq (\epsilon_k^2 + \|z_k - z_{k+1}\|^2) / (2c_k)$ (cost value gap).*

*(b) If both criteria* (A) *and* (B) *are used, and the dual function $-g_0$ in* (A.5) *further satisfies* (QG) *with a coefficient $\mu_q > 0$, then there exists a $\hat{k} > 0$ such that $\text{dist}(z_{k+1}, S) \leq \mu_k \cdot \text{dist}(z_k, S)$, $\forall k \geq \hat{k}$, where $\mu_k = \frac{\theta_k + 2\delta_k}{1 - \delta_k}$ with $\theta_k = 1 / \sqrt{c_k \mu_q + 1} < 1$.*

The first asymptotic convergence for $\lim_{k \to \infty} z_k = z_\infty \in S$ directly comes from Lemma 1, and the asymptotic convergence in terms of primal feasibility and cost value gap is taken from [23, Theorem 4]. The second linear convergence in Lemma A.2 is only guaranteed for the *dual* iterates $z_k$, which is also implied by Lemma 1. However, the rate of the *primal* iterates $x_k$ is unclear even with the quadratic growth condition. With a much stronger condition on the Lagrangian, the classical result in [23, Theorem 5] can ensure linear convergence of $x_k$.

## A.3 Augmented Lagrange functions for inequality constraints

One straightforward way to define the augmented Lagrangian for the inequality form (A.4) is to first transform (A.4) into the following equivalent problem

$$\min_{x \in \mathcal{X}_0, v \geq 0} \quad f(x)$$
$$\text{subject to} \quad f_i(x) + v = 0, \quad \forall i = 1, 2, \ldots, m. \tag{A.8}$$

Then the augmented Lagrangian for the above equality-constrained problem with a penalty term $r > 0$ is defined as

$$L_r(x, v, z) = f(x) + \sum_{i=1}^{m} z_i(f_i(x) + v_i) + \frac{r}{2} \sum_{i=1}^{m} (f_i(x) + v_i)^2$$
$$= f(x) + \frac{1}{2r} \left( \|z + r(f(x) + v)\|^2 - \|z\|^2 \right).$$

Note that $L_r$ is a decoupled quadratic function in terms of $v_i$. Minimizing $L_r(x, v, z)$ with respect to $v \geq 0$ with a fixed $x$ and $y$, we derive the optimal $v$ as

$$v_i = \max \left\{ 0, -\frac{z_i}{r} - f_i(x) \right\}, \ \forall i = 1, \ldots, m.$$

Upon plugging the optimal $v$ in the $L_r$ with some arrangements, we arrive at the classical augmented Lagrangian for the problem with inequality constraints (A.4)

$$L_r(x, z) = f(x) + \frac{1}{2r} \left( \|\Pi_{\mathbb{R}^m_+}(z + rf(x))\|^2 - \|z\|_2 \right).$$

## A.4 Bounded Linear regularity and regularity conditions

We review an important concept: *bounded linear regularity* of a collection of closed convex sets.

**Definition A.2** (**Bounded linear regularity [53, Definition 5.6]**). Let $D_1, \ldots, D_m \subseteq \mathbb{R}^n$ be closed convex set. Suppose $D := D_1 \cap \cdots \cap D_m$ is nonempty. The collection $\{D_1, \ldots, D_m\}$ is called *boundedly linear regular* if for every bounded set $\mathcal{V} \subseteq \mathbb{R}^n$, there exists a constant $\kappa > 0$ such that

$$\text{dist}(x, D) \leq \kappa \max\{\text{dist}(x, D_1), \ldots, \text{dist}(x, D_m)\}, \ \forall x \in \mathcal{V}.$$

Bounded linear regularity allows us to bound (up to some constant) the distance to the intersection $D$ (which can be complicated) by the maximum of the distance to $D_i$, which is often easier to compute. The following result shows a sufficient condition for bounded linear regularity.

**Lemma A.3** ([54, Collorary 3]). *Let $D_1, \ldots, D_m \subseteq \mathbb{R}^n$ be closed convex set. Suppose that $D_1, \ldots, D_r (r \leq m)$ are polyhedra. Then $\{D_1, \ldots, D_m\}$ is boundedly linear regular if*

$$(\cap_{i=1,2,\ldots,r} D_i) \bigcap (\cap_{i=r+1,\ldots,m} \text{relint}(D_i)) \neq \emptyset,$$

*where* relint *denotes the relative interior.*

This result indicates that a collection of closed convex sets is boundedly linear regular if the relative interiors of non-polyhedral sets have a non-empty intersection with the rest of the polyhedra. Lemma A.3 plays a key role in the proof of Theorems 1 and 2. In particular, as we will see in Appendix D.1, the optimal solution set of (P), i.e., $\Omega_P$, can be characterized as

$$\Omega_P = \mathcal{X}_0 \cap (\partial g)^{-1}(-Z^\star), \quad \forall (y^\star, Z^\star) \in \Omega_D,$$

where $\mathcal{X}_0 = \{X \in \mathbb{S}^n_+ \mid \mathcal{A}(X) = b\}$. Thus, assuming dual strict complementarity (there exists $X^\star \in \text{relint}((\partial g)^{-1}(-Z^\star)))$, given a compact set $\mathcal{U} \subseteq \mathbb{S}^n$ containing $X^\star$, Lemma A.3 ensures the following holds

$$\text{dist}(X, \Omega_P) \leq \alpha_1 \text{dist}(X, \mathcal{X}_0) + \alpha_2 \text{dist}(X, (\partial g)^{-1}(-Z^\star)), \ \forall X \in \mathcal{U},$$

where $\alpha_1$ and $\alpha_2$ are two positive constants.

# B  Exact penalty forms of conic optimization

In this section, we characterize the connection between exact penalty functions and indicator functions (6). Most importantly, we give a simple proof for the exact penalty functions. The main result is stated in Proposition B.2. We first state the preimage characterization of subdifferential mapping of the exact penalty function in Proposition B.1, which is a more general result of the first part of Lemma 3 (i.e., (11)). The second part of Lemma 3 (i.e., (12)) is postponed to Appendix C.

## B.1  Subdifferential of the exact penalty function

**Proposition B.1.** *Let* $Z \in \mathbb{S}_+^n$ *and define*

$$l(X) = \rho \max\{0, \lambda_{\max}(-X)\} \text{ with } \rho > \mathbf{tr}(Z). \tag{B.1}$$

*Then for all* $S \in \mathbb{S}_+^n$ *satisfying* $\rho > \mathbf{tr}(S)$*, it holds that*

$$(\partial l)^{-1}(-S) = (\delta_{\mathbb{S}_+^n})^{-1}(-S) = \mathcal{N}_{(\mathbb{S}_+^n)^\circ}(-S). \tag{B.2}$$

One can see that (11) is a special case of Proposition B.1 with $S = \bar{Z}$. We postpone the proof of Proposition B.1 to Appendix B.2. Be aware that (B.2) does not mean the mappings $(\partial l)^{-1}$ and $(\delta_{\mathbb{S}_+^n})^{-1}$ are the same for each point in the domain, while the equality only holds for certain points satisfying the relationship with the penalty coefficient $\rho$ in (B.1). We use a simple example to illustrate this subtlety.

**Example B.1.** *Consider* $l : \mathbb{R} \to \mathbb{R}$ *as* $l(x) = 2\max\{0, -x\}$ *which is the scalar version of the function in* (B.1) *with* $\rho = 2$*. The subdifferential* $\partial l : \mathbb{R} \rightrightarrows \mathbb{R}$ *and* $\partial \delta_{\mathbb{R}_+} = \mathcal{N}_{\mathbb{R}_+}$ *can be computed as*

$$\partial l(x) = \begin{cases} 0, & \text{if } x > 0, \\ [-2, 0], & \text{if } x = 0, \quad \text{and} \quad \partial \delta_{\mathbb{R}_+}(x) = \begin{cases} 0, & \text{if } x > 0, \\ (-\infty, 0], & \text{if } x = 0, \\ \emptyset, & \text{if } x < 0. \end{cases} \\ -2, & \text{if } x < 0, \end{cases}$$

*Thus, the preimages can be verified as*

$$(\partial l)^{-1}(-z) = \begin{cases} \emptyset, & \text{if } z > 2, \\ (-\infty, 0], & \text{if } z = 2, \\ 0, & \text{if } 0 < z < 2, \quad \text{and} \quad \partial(\delta_{\mathbb{R}_+})^{-1}(-z) = \begin{cases} 0, & \text{if } z > 0, \\ [0, \infty), & \text{if } z = 0, \\ \emptyset, & \text{if } z < 0. \end{cases} \\ [0, \infty), & \text{if } z = 0, \\ \emptyset, & \text{if } z < 0, \end{cases}$$

*This shows that* $(\partial l)^{-1}(-z) = (\delta_{\mathbb{R}_+})^{-1}(-z)$ *holds only when* $z < 2$*.*

With the preimage characterization in Proposition B.1, the following result on the exact penalty functions follows easily by choosing the penalty term $\rho$ correctly.

**Proposition B.2** (Exact penalty function). *Assuming strong duality holds for* (P) *and* (D)*,* (4) *is equivalent to the primal SDP* (P) *if the function* $g$ *in* (4) *is chosen to be the corresponding indicator function, i.e.,* $g = \delta_{\mathbb{S}_+^n}$*, or the exact penalty function*

$$l(X) = \rho \max\{0, \lambda_{\max}(-X)\} \text{ with } \rho > \mathbf{tr}(Z^\star), \tag{B.3}$$

*where* $(y^\star, Z^\star) \in \Omega_D$ *is an optimal dual solution of* (D)*.*

*Proof.* The case for the indicator function $\delta_{\mathbb{S}_+^n}$ is clear. We consider the case for the exact penalty function. Note the dual of (4) is

$$\begin{aligned} \max_{y,Z} \quad & \langle b, y \rangle \\ \text{subject to} \quad & \mathcal{A}^*(y) + Z = C, \\ & \mathbf{tr}(Z) \leq \rho, \\ & Z \in \mathbb{S}_+^n. \end{aligned} \tag{B.4}$$

Let $\Omega_{P_2}$ and $\Omega_{D_2}$ denote the optimal solution of (4) and (B.4) respectively. First, it is clear that $\Omega_{D_2} \subseteq \Omega_D$ as there is an extra constraint in (B.4), compared with (D). Second, as the problem is convex, a pair of primal and dual solutions $(\hat{X}^\star, \hat{y}^\star, \hat{Z}^\star)$ solves (4) and (B.4) if and only if the following KKT system holds

$$\mathcal{A}(\hat{X}^\star) = b, \ Z^\star = C - \mathcal{A}^*(\hat{y}^\star), \ 0 \in \hat{Z}^\star + \partial g(\hat{X}^\star).$$

Then the primal solution set can be written as

$$\Omega_{P_2} = \mathcal{X}_0 \cap (\partial g)^{-1}(-\hat{Z}^\star), \quad \forall (\hat{y}^\star, \hat{Z}^\star) \in \Omega_{D_2}.$$

where $\mathcal{X}_0 = \{X \in \mathbb{S}^n \mid \mathcal{A}(X) = b\}$. Note that $(y^\star, Z^\star)$ is an optimal solution to (B.4) as $(y^\star, Z^\star)$ is feasible in (B.4) and $\langle b, y^\star \rangle \leq \max_{y,Z}$ (B.4) $\leq d^\star = \langle b, y^\star \rangle$ by construction. We can write $\Omega_{P_2}$ as

$$\Omega_{P_2} = \mathcal{X}_0 \cap (\partial g)^{-1}(-Z^\star) = \mathcal{X}_0 \cap (\partial \delta_{\mathbb{S}^n_+})^{-1}(-Z^\star),$$

where the last equality uses $(\partial g)^{-1}(-Z^\star) = (\partial \delta_{\mathbb{S}^n_+})^{-1}(-Z^\star)$ in (B.2) as $\mathbf{tr}(Z^\star) < \rho$ and $Z^\star \in \mathbb{S}^n_+$. On the other hand, we can write $\Omega_P$ as

$$\Omega_P = \mathcal{X}_0 \cap (\partial \delta_{\mathbb{S}^n_+})^{-1}(-Z^\star).$$

It is clear that $\Omega_{P_2} = \Omega_P$ and the proof is complete. $\qquad \square$

*Remark* B.1 (Dual problem). Note that Proposition B.2 only guarantees the cost value and solution sets of (4) and (P) to be the same, while the solution sets of (D) and the dual of (4) may differ, as seen in the proof of Proposition B.2. To fully equate (D) and the dual of (4), one can choose the penalty coefficient $\rho$ as $\rho > \sup_{(y^\star, Z^\star) \in \Omega_D} \mathbf{tr}(Z^\star)$. This choice of $\rho$ can guarantee every optimal solution in (D) is also an optimal solution in (B.4) as the extra constraint in (B.4) (compared with (D)) does not affect the solution set. Combining this with the fact $\Omega_{D_2} \subseteq \Omega_D$, we conclude that $\Omega_D = \Omega_{D_2}$.

## B.2 Proof for Proposition B.1

The following proposition showcases the subdifferential calculation of the function $l(X) = \rho \max\{0, \lambda_{\max}(-X)\}$ with $\rho \geq 0$. It is a standard result of the subdifferential of the maximal eigenvalue of symmetric matrices and standard subdifferential calculus.

**Proposition B.3** ([55, Theorem 2] and [45, Theorem 2.87]). *Let $\rho \geq 0$ and $X \in \mathbb{S}^n$ with $\lambda_{\max}(-X)$ having $t$ multiplicity. The subdifferential of $l(X) = \rho \max\{0, \lambda_{\max}(-X)\}$ is characterized by*

$$\partial l(X) = \begin{cases} 0 & \text{if } \lambda_{\min}(X) > 0, \\ \{-\rho P S P^\mathsf{T} \mid S \in \mathbb{S}^t_+, \mathbf{tr}(S) \leq 1\} & \text{if } \lambda_{\min}(X) = 0, \\ \{-\rho P S P^\mathsf{T} \mid S \in \mathbb{S}^t_+, \mathbf{tr}(S) = 1\} & \text{if } \lambda_{\min}(X) < 0, \end{cases} \tag{B.5}$$

*where columns of $P \in \mathbb{R}^{n \times t}$ are orthonormal eigenvectors corresponding to $\lambda_{\max}(-X)$.*

**Proposition B.4.** *Let $Z \in \mathbb{S}^n_+ \setminus \{0\}$ and write $Z = \begin{bmatrix} P_1 & P_2 \end{bmatrix} \begin{bmatrix} \Lambda_1 & 0 \\ 0 & 0 \end{bmatrix} \begin{bmatrix} P_1^\mathsf{T} \\ P_2^\mathsf{T} \end{bmatrix}$ with $\Lambda_1 \in \mathbb{S}^r_{++}$. Let $l(X) = \rho \max\{0, \lambda_{\max}(-X)\}$ with $\rho > \mathbf{tr}(Z)$. We have*

$$(\partial l)^{-1}(0) = \mathbb{S}^n_+,$$
$$(\partial l)^{-1}(-Z) = (\mathcal{N}_{\mathbb{S}^n_+})^{-1}(-Z).$$

*Proof.* The case $(\partial l)^{-1}(0) = \mathbb{S}^n_+$ can be easily observed from the first two cases in (B.5). We then consider the case $Z \in \mathbb{S}^n_+ \setminus \{0\}$. Given a $X \in \mathbb{S}^n$, we use the matrix $P_X \in \mathbb{R}^{n \times t}$ to denote the orthonormal eigenvectors corresponding to $\lambda_{\max}(-X)$ where $t$ is the multiplicity of $\lambda_{\max}(-X)$. It follows that

$$\begin{aligned} (\partial l)^{-1}(-Z) &= \{X \in \mathbb{S}^n \mid -Z \in \partial l(X)\} \\ &\overset{(a)}{=} \{X \in \mathbb{S}^n_+ \setminus \mathbb{S}^n_{++} \mid Z \in \rho P_X S P_X^\mathsf{T}, S \in \mathbb{S}^t_+, \mathbf{tr}(S) < 1\} \\ &\overset{(b)}{=} \left\{ \begin{bmatrix} P_1 & P_2 \end{bmatrix} \begin{bmatrix} 0 & 0 \\ 0 & B \end{bmatrix} \begin{bmatrix} P_1^\mathsf{T} \\ P_2^\mathsf{T} \end{bmatrix} \mid 0 \in \mathbb{R}^{r \times r}, B \in \mathbb{S}^{n-r}_+ \right\} \\ &\overset{(c)}{=} \mathcal{N}_{\mathbb{S}^n_-}(-Z) \\ &= (\mathcal{N}_{\mathbb{S}^n_+})^{-1}(-Z), \end{aligned}$$

where the domain $\mathbb{S}^n_+ \setminus \mathbb{S}^n_{++}$ and the strict inequality $\mathbf{tr}(S) < 1$ in $(a)$ is due to the choice $\rho > \mathbf{tr}(Z)$, and $(c)$ is a common form of $\mathcal{N}_{\mathbb{S}^n_-}(-Z)$ [28, Complementary face in Section 3.3]. We argue $(b)$ is correct by verifying the following two sets are equivalent.

$$D_1 = \{X \in \mathbb{S}^n_+ \setminus \mathbb{S}^n_{++} \mid Z \in \rho P_X S P_X^\mathsf{T}, S \in \mathbb{S}^t_+, \mathbf{tr}(S) < 1\},$$

$$D_2 = \left\{ [P_1 \quad P_2] \begin{bmatrix} 0 & 0 \\ 0 & B \end{bmatrix} \begin{bmatrix} P_1^\mathsf{T} \\ P_2^\mathsf{T} \end{bmatrix} \mid 0 \in \mathbb{R}^{r \times r}, B \in \mathbb{S}^{n-r}_+ \right\}.$$

- $(D_1) \Rightarrow (D_2)$ : Let $X \in D_1$. Write $X = P_\alpha 0 P_\alpha^\mathsf{T} + P_\beta \Lambda_\beta P_\beta^\mathsf{T}$ with $P_\alpha \in \mathbb{R}^{n \times t}$ and $P_\beta \in \mathbb{R}^{n \times (n-t)}$. As $\mathrm{span}(P_1) \subseteq \mathrm{span}(P_\alpha)$, we have $\mathrm{span}(P_\beta) \subseteq \mathrm{span}(P_2)$. Thus, there exists $U \in \mathbb{R}^{n \times (n-r)}$ such that $U^\mathsf{T} U = I_{n-r}$ and $P_\beta = P_2 U$. We then have

$$P_\beta \Lambda_\beta P_\beta^\mathsf{T} = P_2 (U \Lambda_\beta U^\mathsf{T}) P_2^\mathsf{T}.$$

This means $X \in D_2$.
- $(D_2) \Rightarrow (D_1)$ : Let $X \in D_2$. It is clear that $\mathrm{span}(P_1)$ is in the eigenspace of $0$ of the matrix $X$. Choosing $S = \frac{\Lambda_1}{\rho} \in \mathbb{S}^r_{++}$, we can recover $Z$ as

$$\rho P_1 \frac{\Lambda_1}{\rho} P_1^\mathsf{T} = Z.$$

Also, $\mathbf{tr}(S) = \mathbf{tr}(\Lambda_1)/\rho < 1$, so we conclude $X \in D_1$. $\qquad \square$

## C   Growth properties of indicator and exact penalty functions in Lemma 3

One key step in the proof of Theorems 1 and 2 is the growth properties of the indicator function or the penalty function (10), summarized in (12) in Lemma 3. In this section, we complete its proof details. We first establish a lower bound of $\langle \bar{Z}, X \rangle$ in terms of the distance $\mathrm{dist}(X, \mathcal{N}_{(\mathbb{S}^n_+)^\circ}(-\bar{Z}))$. Our proof uses some techniques in [27].

**Lemma C.1.** *Let $\bar{X}, \bar{Z} \in \mathbb{S}^n_+$ satisfying $\langle \bar{X}, \bar{Z} \rangle = 0$ (i.e., both $\bar{X}$ and $\bar{Z}$ are on the boundary unless one of them is zero). For any positive $\mu \in (0, \infty)$, there exists a constant $\kappa > 0$ such that*

$$\langle \bar{Z}, X \rangle \geq \kappa \cdot \mathrm{dist}^2 \left( X, \mathcal{N}_{(\mathbb{S}^n_+)^\circ}(-\bar{Z}) \right), \quad \forall X \in \mathbb{S}^n_+ \cap \mathbb{B}(\bar{X}, \mu), \tag{C.1}$$

*where $\mathbb{B}(\bar{X}, \mu) = \{X \in \mathbb{S}^n \mid \|X - \bar{X}\| \leq \mu\}$ denotes a closed ball with radius $\mu$ and center $\bar{X}$.*

Note that (C.1) is the reformulation of

$$0 \geq 0 - \langle \bar{Z}, X \rangle + \kappa \cdot \mathrm{dist}^2 \left( X, \mathcal{N}_{(\mathbb{S}^n_+)^\circ}(-\bar{Z}) \right), \quad \forall X \in \mathbb{S}^n_+ \cap \mathbb{B}(\bar{X}, \mu). \tag{C.2}$$

Indeed, (C.2) is the same as (12) in Lemma 3 when choosing $l = \delta_{\mathbb{S}^n_+}$ (recall that we have $\delta_{\mathbb{S}^n_+}(\bar{X}) = 0$, and the relationship in (11)). Therefore, Lemma 3 for the indicator function $l = \delta_{\mathbb{S}^n_+}$ is a direct consequence of Lemma C.1. The proof of Lemma C.1 is given in Appendix C.1. In Appendix C.2, we will use Lemma C.1 to prove the growth property for the exact penalty function $l$ in (10).

### C.1   Proof of Lemma C.1

We only need to consider the case where $\bar{Z} \neq 0$, since the case $\bar{Z} = 0$ is true by observing that

$$\langle X, \bar{Z} \rangle = 0, \; \mathrm{dist}(X, \mathcal{N}_{(\mathbb{S}^n_+)^\circ}(0)) = \mathrm{dist}(X, \mathbb{S}^n_+) = 0, \quad \forall X \in \mathbb{S}^n_+ \cap \mathbb{B}(\bar{X}, \mu).$$

We first introduce the following useful inequality for positive semidefinite matrices.

**Lemma C.2.** *Suppose $\begin{bmatrix} A & B \\ B^\mathsf{T} & D \end{bmatrix} \in \mathbb{S}^n_+$. Then $\|D\|_{\mathrm{op}} \, \mathbf{tr}(A) \geq \|B\|^2$.*

We now proceed with the proof. Fix a $\mu > 0$. Consider $\bar{X}, \bar{Z} \in \mathbb{S}^n_+$ and $\langle \bar{X}, \bar{Z} \rangle = 0$. Let $X \in \mathbb{B}(\bar{X}, \mu) \cap \mathbb{S}^n_+$ and write

$$\bar{Z} = [P_1 \quad P_2] \begin{bmatrix} \Lambda_1 & 0 \\ 0 & 0 \end{bmatrix} \begin{bmatrix} P_1^\mathsf{T} \\ P_2^\mathsf{T} \end{bmatrix},$$

where $\Lambda_1 \in \mathbb{S}_+^r$ is a diagonal matrix containing the positive eigenvalues of $\bar{Z}$. Let $P = [P_1 \ P_2]$ be the orthonormal eigenvectors matrix. The projection of $X$ onto $\mathcal{N}_{(\mathbb{S}_+^n)^\circ}(-\bar{Z})$ can be verified as

$$P \begin{bmatrix} 0 & 0 \\ 0 & P_2^\mathsf{T} X P_2 \end{bmatrix} P^\mathsf{T} = \underset{Y \in \mathcal{N}_{(\mathbb{S}_+^n)^\circ}(-\bar{Z})}{\operatorname{argmin}} \|X - Y\|^2.$$

Hence, the distance $\operatorname{dist}^2(X, \mathcal{N}_{(\mathbb{S}_+^n)^\circ}(-\bar{Z}))$ can be computed as

$$\begin{aligned} \operatorname{dist}^2(X, \mathcal{N}_{(\mathbb{S}_+^n)^\circ}(-\bar{Z})) &= \left\| X - P \begin{bmatrix} 0 & 0 \\ 0 & P_2^\mathsf{T} X P_2 \end{bmatrix} P^\mathsf{T} \right\|^2 \\ &= \left\| \begin{bmatrix} P_1^\mathsf{T} X P_1 & P_1^\mathsf{T} X P_2 \\ P_2^\mathsf{T} X P_1 & 0 \end{bmatrix} \right\|^2 = \|P_1^\mathsf{T} X P_1\|^2 + 2\|P_1^\mathsf{T} X P_2\|^2, \end{aligned} \tag{C.3}$$

where the second equality uses the unitary invariance. It remains to estimate the terms $\|P_1^\mathsf{T} X P_1\|^2$ and $\|P_1^\mathsf{T} X P_2\|^2$.

By the assumption $X \in \mathbb{B}(\bar{X}, \mu) \cap \mathbb{S}_+^n$ and unitary invariance, we have

$$\begin{aligned} \mu \geq \|X - \bar{X}\| = \|P^\mathsf{T}(X - \bar{X})P\| &\geq \max\{\|P_1^\mathsf{T}(X - \bar{X})P_1\|, \|P_2^\mathsf{T}(X - \bar{X})P_2\|\} \\ &\geq \max\{\|P_1^\mathsf{T} X P_1\|_{\mathrm{op}}, \|P_2^\mathsf{T}(X - \bar{X})P_2\|_{\mathrm{op}}\}, \end{aligned} \tag{C.4}$$

where the last inequality uses the fact that $P_1^\mathsf{T} \bar{X} P_1 = 0$ as $\langle \bar{X}, \bar{Z} \rangle = 0$ and $\bar{X}, \bar{Z} \in \mathbb{S}_+^n$, and the relationship $\|\cdot\| \geq \|\cdot\|_{\mathrm{op}}$. It follows that

$$\|P_1^\mathsf{T} X P_1\|^2 = \langle P_1^\mathsf{T} X P_1, P_1^\mathsf{T} X P_1 \rangle \leq \|P_1^\mathsf{T} X P_1\|_{\mathrm{op}} \|P_1^\mathsf{T} X P_1\|_* \leq \mu \cdot \mathbf{tr}(P_1^\mathsf{T} X P_1), \tag{C.5}$$

where the second inequality uses the generalized Cauchy-Schwarz inequality ($\|\cdot\|_*$ and $\|\cdot\|_{\mathrm{op}}$ norms are dual to each other; we have $|\langle X, Z \rangle| \leq \|X\|_* \|Z\|_{\mathrm{op}}, \forall X, Z \in \mathbb{S}^n$), and the last inequality uses (C.4) and $\|P_1^\mathsf{T} X P_1\|_* = \mathbf{tr}(P_1^\mathsf{T} X P_1)$ as $X \in \mathbb{S}_+^n$.

On the other hand, applying Lemma C.2 on $P^\mathsf{T} X P$ yields

$$\|P_1^\mathsf{T} X P_2\|^2 \leq \|P_2^\mathsf{T} X P_2\|_{\mathrm{op}} \mathbf{tr}(P_1^\mathsf{T} X P_1) \leq (\mu + \|\bar{X}\|) \cdot \mathbf{tr}(P_1^\mathsf{T} X P_1), \tag{C.6}$$

where the last inequality is due to (C.4). Moreover, we have

$$\langle \bar{Z}, X \rangle = \langle P_1 \Lambda_1 P_1^\mathsf{T}, X \rangle = \langle \Lambda_1, P_1^\mathsf{T} X P_1 \rangle \geq \lambda_{\min}(\Lambda_1) \mathbf{tr}(P_1^\mathsf{T} X P_1), \tag{C.7}$$

where $\lambda_{\min}(\Lambda_1) > 0$. Putting (C.3), (C.5) and (C.6) together and choosing $\kappa = \frac{\lambda_{\min}(\Lambda_1)}{3\mu + 2\|\bar{X}\|}$ gives us

$$\kappa \cdot \operatorname{dist}^2(X, \mathcal{N}_{(\mathbb{S}_+^n)^\circ}(-\bar{Z})) \leq \kappa(3\mu + 2\|\bar{X}\|) \mathbf{tr}(P_1^\mathsf{T} X P_1) \leq \langle \bar{Z}, X \rangle,$$

where the last inequality is due to (C.7). This completes the proof.

### C.2 Growth property of the exact penalty function $l$ in (10)

Throughout this section, we fix a pair of matrices $(\bar{X}, \bar{Z}) \in \mathbb{S}_+^n \times \mathbb{S}_+^n$ satisfying $\langle \bar{X}, \bar{Z} \rangle = 0$. We need to prove for any positive value $\mu \in (0, \infty)$, there exists a positive constant $\kappa > 0$ such that

$$l(X) \geq l(\bar{X}) + \langle -\bar{Z}, X - \bar{X} \rangle + \kappa \cdot \operatorname{dist}^2\left(X, (\partial l)^{-1}(-\bar{Z})\right), \quad \forall X \in \mathbb{B}(\bar{X}, \mu). \tag{C.8}$$

Note that it is equivalent to showing that there exists a $\kappa > 0$ such that

$$l(X) \geq \underbrace{\langle -\bar{Z}, X \rangle}_{R_1} + \kappa \cdot \underbrace{\operatorname{dist}^p\left(X, \mathcal{N}_{(\mathbb{S}_+^n)^\circ}(-\bar{Z})\right)}_{R_2}, \quad \forall X \in \mathbb{B}(\bar{X}, \mu) \tag{C.9}$$

as $l(\bar{X}) = 0$ and $\langle \bar{Z}, \bar{X} \rangle = 0$ by assumption. This result (C.9) has already applied a non-trivial fact $(\partial l)^{-1}(-\bar{Z}) = (\mathcal{N}_{\mathbb{S}_+^n})^{-1}(-\bar{Z}) = \mathcal{N}_{(\mathbb{S}_+^n)^\circ}(-\bar{Z})$ for the exact penalty function (10), which has been established in Proposition B.1.

In the following, we will bound the terms $R_1$ and $R_2$ in (C.9). The term $R_2$ will rely on Lemma C.1. One key difference between (C.9) and (C.2) is that the penalty function $l(X) < \infty$ when $X \notin \mathbb{S}^n_+$, while $\delta_{\mathbb{S}^n_+}(X) = \infty, \forall X \notin \mathbb{S}^n_+$.

Let us fix a $\mu > 0$, consider $X \in \mathbb{B}(\bar{X}, \mu)$, and write $\rho = \delta + \mathbf{tr}(\bar{Z})$. We first consider the case when $\bar{Z} = 0$. Note that $\langle \bar{Z}, X \rangle = 0$ and $\mathcal{N}_{(\mathbb{S}^n_+)^\circ}(0) = \mathbb{S}^n_+$. The distance $\text{dist}(X, \mathcal{N}_{(\mathbb{S}^n_+)^\circ}(0))$ can be upper bounded as

$$\text{dist}^2(X, \mathcal{N}_{(\mathbb{S}^n_+)^\circ}(0)) = \|X - \Pi_{\mathbb{S}^n_+}(X)\|^2 \le n \max\{0, \lambda_{\max}(-X)\}^2 \le n\mu \max\{0, \lambda_{\max}(-X)\}.$$

Choosing $\kappa = \frac{\rho}{n\mu}$ yields

$$\langle -Z^\star, X \rangle + \kappa \cdot \text{dist}^2(X, \mathcal{N}_{(\mathbb{S}^n_+)^\circ}(0)) \le \rho \max\{0, \lambda_{\max}(-X)\} = l(X), \quad \forall X \in \mathbb{B}(\bar{X}, \mu).$$

We then consider the case $\bar{Z} \ne 0$.

- We first bound $R_1 = \langle -\bar{Z}, X \rangle$:

$$\begin{aligned}
\langle -\bar{Z}, X \rangle &= \left\langle \bar{Z}, \Pi_{\mathbb{S}^n_+}(X) - X \right\rangle - \left\langle \bar{Z}, \Pi_{\mathbb{S}^n_+}(X) \right\rangle \\
&\le \|\bar{Z}\|_* \|\Pi_{\mathbb{S}^n_+}(X) - X\|_{\text{op}} - \left\langle \bar{Z}, \Pi_{\mathbb{S}^n_+}(X) \right\rangle \\
&= \mathbf{tr}(\bar{Z}) \max\{0, \lambda_{\max}(-X)\} - \left\langle \bar{Z}, \Pi_{\mathbb{S}^n_+}(X) \right\rangle,
\end{aligned} \tag{C.10}$$

where the inequality applies the general Cauchy-Schwarz inequality ($\|\cdot\|_*$ and $\|\cdot\|_{\text{op}}$ norms are dual to each other; we have $|\langle X, Z \rangle| \le \|X\|_* \|Z\|_{\text{op}}, \forall X, Z \in \mathbb{S}^n)$.

- We then bound $R_2 = \text{dist}^2(X, \mathcal{N}_{(\mathbb{S}^n_+)^\circ}(-\bar{Z}))$: Let $Y$ be the projection of $\Pi_{\mathbb{S}^n_+}(X)$ on to $\mathcal{N}_{(\mathbb{S}^n_+)^\circ}(-\bar{Z})$, i.e., $Y = \text{argmin}_{V \in \mathcal{N}_{(\mathbb{S}^n_+)^\circ}(-\bar{Z})} \|\Pi_{\mathbb{S}^n_+}(X) - V\|$. We then have

$$\text{dist}^2\left(X, \mathcal{N}_{(\mathbb{S}^n_+)^\circ}(-\bar{Z})\right) \le \|X - Y\|^2 \le 2\|X - \Pi_{\mathbb{S}^n_+}(X)\|^2 + 2\|\Pi_{\mathbb{S}^n_+}(X) - Y\|^2. \tag{C.11}$$

The first term $\|X - \Pi_{\mathbb{S}^n_+}(X)\|^2$ can be bounded by

$$\|X - \Pi_{\mathbb{S}^n_+}(X)\|^2 \le n \max\{0, \lambda_{\max}(-X)\}^2 \le n\mu \max\{0, \lambda_{\max}(-X)\},$$

where the last inequality comes from $|\lambda_{\min}(X) - \lambda_{\min}(\bar{X})| = |\lambda_{\max}(-X)| \le \mu$ as $\lambda_{\min}(\bar{X}) = 0$ and $\|X - \bar{X}\| \le \mu$.

The second term $\|\Pi_{\mathbb{S}^n_+}(X) - Y\|^2$ can be bounded by Lemma C.1, i.e., there is a $\kappa' > 0$ such that

$$\|\Pi_{\mathbb{S}^n_+}(X) - Y\|^2 \le \frac{1}{\kappa'} \left\langle \Pi_{\mathbb{S}^n_+}(X), \bar{Z} \right\rangle.$$

Putting everything together and choosing $\kappa = \min\{\frac{\delta}{2n\mu}, \frac{\kappa'}{2}\}$ yields

$$\begin{aligned}
&\langle -\bar{Z}, X \rangle + \kappa \cdot \text{dist}^2\left(X, \mathcal{N}_{(\mathbb{S}^n_+)^\circ}(-\bar{Z})\right) \\
&\le \mathbf{tr}(\bar{Z}) \max\{0, \lambda_{\max}(-X)\} - \left\langle \bar{Z}, \Pi_{\mathbb{S}^n_+}(X) \right\rangle \\
&\quad + \kappa 2n\mu \max\{0, \lambda_{\max}(-X)\} + \frac{2\kappa}{\kappa'} \cdot \left\langle \Pi_{\mathbb{S}^n_+}(X), \bar{Z} \right\rangle \\
&\le \mathbf{tr}(\bar{Z}) \max\{0, \lambda_{\max}(-X)\} + \delta \max\{0, \lambda_{\max}(-X)\} \\
&= \rho \max\{0, \lambda_{\max}(-X)\} = l(X), \quad \forall X \in \mathbb{B}(\bar{X}, \mu).
\end{aligned}$$

## D   Growth growth under strict complementarity

As mentioned in the main text, the key step to prove Theorems 1 and 2 is to establish the growth property (12) in Lemma 3 which we have proved in Appendix C. Once Lemma 3 is established, with strict complementarity assumption, we can deduce Proposition 1. In this appendix, we first provide proof for Proposition 1 in Appendix D.1 and use a simple example to illustrate the quadratic growth property in Appendix D.2.

## D.1 Proof of Proposition 1

The following proof is adapted from [27]. We first prove the primal case. Fix a $\mu > 0$ and $X^\star \in \Omega_{\mathrm{P}}$. As the problem is convex, a pair of primal and dual solutions $(\hat{X}^\star, \hat{y}^\star, \hat{Z}^\star)$ solves (4) and its dual problem if and only if the following KKT system holds

$$\mathcal{A}(\hat{X}^\star) = b, \ \hat{Z}^\star = C - \mathcal{A}^*(\hat{y}^\star), \ 0 \in \hat{Z}^\star + \partial g(\hat{X}^\star).$$

Combined with the fact that any primal and dual solution forms a pair, the optimal primal solution $\Omega_{\mathrm{P}}$ can be characterized as

$$\Omega_{\mathrm{P}} = \mathcal{X}_0 \cap (\partial g)^{-1}(-\hat{Z}^\star), \quad \forall (\hat{y}^\star, \hat{Z}^\star) \in \Omega_{\mathrm{D}},$$

where $\mathcal{X}_0 = \{X \in \mathbb{S}^n \mid \mathcal{A}(X) = b\}$. Let $(\bar{X}^\star, \bar{y}^\star, \bar{Z}^\star)$ be a pair of primal and dual solutions satisfying $\bar{X}^\star \in \mathrm{relint}((\partial g)^{-1}(-\bar{Z}^\star))$ (strict complementarity assumption). There exist $\alpha_1, \kappa_1, \kappa_2 > 0$ such that for all $X \in \mathbb{B}(X^\star, \mu)$ the following holds

$$
\begin{aligned}
\mathrm{dist}^2(X, \Omega_{\mathrm{P}}) &= \mathrm{dist}^2(X, \mathcal{X}_0 \cap (\partial g)^{-1}(-\bar{Z}^\star)) \\
&\leq \alpha_1 (\mathrm{dist}(X, \mathcal{X}_0) + \mathrm{dist}(X, (\partial g)^{-1}(-\bar{Z}^\star)))^2 \\
&\leq \kappa_1 (\mathrm{dist}^2(X, \mathcal{X}_0) + \mathrm{dist}^2(X, (\partial g)^{-1}(-\bar{Z}^\star))) \\
&\leq \kappa_2 (\|\mathcal{A}(X) - b\|^2 + \mathrm{dist}^2(X, (\partial g)^{-1}(-\bar{Z}^\star))),
\end{aligned}
\tag{D.1}
$$

where the first inequality applies Lemma A.3, the second inequality applies $(A + B)^2 \leq 2A^2 + 2B^2$ for all $A, B \geq 0$, the third inequality uses the fact that $\mathcal{X}_0$ is an affine space. On the other hand, we know $\left\langle \hat{X}^\star, \bar{Z}^\star \right\rangle = 0$ for all $\hat{X}^\star \in \Omega_{\mathrm{P}}$ naturally, so Lemma 3 can be applied here on $Z^\star$ and $\bar{Z}^\star$ (the proof of Lemma 3 is provided in Appendix C.2). For all $X \in \mathbb{B}(X^\star, \mu)$ it holds that

$$
\begin{aligned}
f_{\mathrm{P}}(X) &= \langle C, X \rangle + g(X) \\
&\geq \langle C, X \rangle + g(X^\star) + \langle \mathcal{A}^*(\bar{y}^\star) - C, X - X^\star \rangle + \kappa \cdot \mathrm{dist}^2(X, (\partial g)^{-1}(-\bar{Z}^\star)) \\
&\geq \langle C, X \rangle + g(X^\star) + \langle \mathcal{A}^*(\bar{y}^\star) - C, X - X^\star \rangle + \frac{\kappa}{\kappa_2} \mathrm{dist}^2(X, \Omega_{\mathrm{P}}) - \kappa \|\mathcal{A}(X) - b\|^2 \\
&\geq f_{\mathrm{P}}(X^\star) - \|\bar{y}^\star\| \|\mathcal{A}(X) - b\| - \kappa \|\mathcal{A}(X) - b\|^2 + \frac{\kappa}{\kappa_2} \cdot \mathrm{dist}^2(X, \Omega_{\mathrm{P}}) \\
&= f_{\mathrm{P}}(X^\star) - (\|\bar{y}^\star\| + \kappa \|\mathcal{A}(X) - b\|) \|\mathcal{A}(X) - b\| + \frac{\kappa}{\kappa_2} \cdot \mathrm{dist}^2(X, \Omega_{\mathrm{P}}) \\
&\geq f_{\mathrm{P}}(X^\star) - (\|\bar{y}^\star\| + \gamma) \|\mathcal{A}(X) - b\| + \frac{\kappa}{\kappa_2} \cdot \mathrm{dist}^2(X, \Omega_{\mathrm{P}}),
\end{aligned}
$$

where the first inequality uses the growth property on $g$ for $X^\star$ and $\bar{Z}^\star$ (12), the second inequality uses (D.1), the third inequality uses the definition of the adjoint operator and Cauchy–Schwarz inequality, and the last inequality uses the fact that $X$ is in a bounded set $\mathbb{B}(X^\star, \mu)$ so there exists some $\gamma \geq 0$ such that $\kappa \|\mathcal{A}(X) - b\| \leq \gamma$. This completes the proof.

We move on to prove the dual case. Similarly, we let $(y^\star, Z^\star) \in \Omega_{\mathrm{D}}$ and $\mu > 0$ and characterize the optimal dual solution as

$$\Omega_{\mathrm{D}} = \mathcal{Z}_0 \cap (\mathbb{R}^m \times (\partial h)^{-1}(-\hat{X}^\star)), \quad \forall \hat{X}^\star \in \Omega_{\mathrm{P}},$$

where $\mathcal{Z}_0 = \{(y, Z) \in \mathbb{R}^m \times \mathbb{S}^n \mid C - \mathcal{A}^*(y) = Z\}$. Let $(\bar{X}^\star, \bar{y}^\star, \bar{Z}^\star)$ be a pair of primal and dual solution satisfying $\bar{Z}^\star \in \mathrm{relint}((\partial h)^{-1}(-\bar{X}^\star))$. Note that

$$\mathrm{relint}(\mathbb{R}^m \times (\partial h)^{-1}(-\bar{X}^\star)) = \mathbb{R}^m \times \mathrm{relint}((\partial h)^{-1}(-\bar{X}^\star)).$$

Therefore, the existence of such a pair of strict complementarity solution implies

$$\mathcal{Z}_0 \cap \mathrm{relint}(\mathbb{R}^m \times (\partial h)^{-1}(-\bar{X}^\star)) \neq \emptyset.$$

Hence, following the same argument to (D.1), we know there exists a constant $\kappa_3 > 0$ such that

$$\mathrm{dist}^2((y, Z), \Omega_{\mathrm{D}}) \leq \kappa_3 (\|C - \mathcal{A}^*(y) - Z\|^2 + \mathrm{dist}^2((y, Z), (\partial h)^{-1}(-\bar{X}^\star))) \tag{D.2}$$

for all $(y, Z) \in \mathbb{B}((y^\star, Z^\star), \mu)$. Also, as $\langle \bar{X}^\star, Z^\star \rangle = 0$, we have

$$
\begin{aligned}
f_{\mathrm{D}}(y, Z) &= -\langle b, y \rangle + h(Z) \\
&\geq -\langle b, y \rangle + h(Z^\star) + \langle -X^\star, Z - Z^\star \rangle + \kappa \cdot \mathrm{dist}^2((y, Z), (\partial h)^{-1}(-\bar{X}^\star)) \\
&= h(Z^\star) + \langle -\bar{X}^\star, \mathcal{A}^*(y) - Z - C + \mathcal{A}^*(y^\star) \rangle + \kappa \cdot \mathrm{dist}^2((y, Z), (\partial h)^{-1}(-\bar{X}^\star)) \\
&\geq f_{\mathrm{D}}(y^\star, Z^\star) - \|\bar{X}^\star\| \|C - \mathcal{A}^*(y) - Z\| - \kappa \|C - \mathcal{A}^*(y) - Z\|^2 + \frac{\kappa}{\kappa_3} \cdot \mathrm{dist}^2((y, Z), \Omega_{\mathrm{D}}) \\
&= f_{\mathrm{D}}(y^\star, Z^\star) - (\|\bar{X}^\star\| + \kappa\|C - \mathcal{A}^*(y) - Z\|)\|C - \mathcal{A}^*(y) - Z\| + \frac{\kappa}{\kappa_3} \cdot \mathrm{dist}^2((y, Z), \Omega_{\mathrm{D}}) \\
&\geq f_{\mathrm{D}}(y^\star, Z^\star) - (\|\bar{X}^\star\| + \gamma)\|C - \mathcal{A}^*(y) - Z\| + \frac{\kappa}{\kappa_3} \cdot \mathrm{dist}^2((y, Z), \Omega_{\mathrm{D}}),
\end{aligned}
$$

where the first inequality uses the growth property on $h$, the second equality rewrites $\langle b, y \rangle = \langle \bar{X}^\star, \mathcal{A}^*(y) \rangle$ and $Z^\star = C - \mathcal{A}^*(y^\star)$, and the last inequality uses the fact that $(y, Z)$ is in a bounded set $\mathbb{B}((y^\star, Z^\star), \mu)$ so there exists a $\gamma \geq 0$ such that $\kappa \|C - \mathcal{A}^*(y) - Z\| \leq \gamma$.

### D.2 Illustration of quadratic growth

In this subsection, we use a simple example to illustrate the quadratic growth property discussed in (9). In particular, we choose the exact penalty formulation for dual SDP (5) (the proof for the exact penalty function can be found in Appendix B). Moreover, the following example shows that even a simple $2 \times 2$ SDP *can not* have *sharp* growth in the objective function.

**Example D.1.** *Consider a SDP* (P) *and* (D) *with the problem data*

$$
C = \begin{bmatrix} 1 & -1 \\ -1 & 1 \end{bmatrix}, A_1 = \begin{bmatrix} 1 & 0 \\ 0 & 0 \end{bmatrix}, A_2 = \begin{bmatrix} 0 & 0 \\ 0 & 1 \end{bmatrix}, \text{ and } b = \begin{bmatrix} 1 \\ 1 \end{bmatrix}.
$$

*It can be verified that both primal and dual SDP have a unique solution*

$$
X^\star = \begin{bmatrix} 1 & 1 \\ 1 & 1 \end{bmatrix}, y^\star = \begin{bmatrix} 0 \\ 0 \end{bmatrix}, \text{ and } Z^\star = \begin{bmatrix} 1 & -1 \\ -1 & 1 \end{bmatrix}.
$$

*Therefore, the optimal cost value is 0. Moreover, it also satisfies strict complementarity as $\mathrm{rank}(X^\star) + \mathrm{rank}(Z^\star) = 1 + 1 = 2$ (see Definition A.1).* (8b) *and* (9b) *then ensure the following exact penalized formulation having quadratic growth property*

$$
f(y) := -b^\mathsf{T} y + \rho \max\{0, \lambda_{\max}(\mathcal{A}^*(y) - C)\},
$$

*where $\rho$ can be chosen as any number satisfying $\rho > \mathbf{tr}(Z^\star) = 2$. Let $S = \{(0, 0)\}$ be the optimal dual solution set and $f^\star = 0$ be the optimal cost value. Figure 2a shows the landscape of the function $f$ with $\rho = 4$ and verifies that the function value $f(y)$ is growing at least quadratically away from the distance to the solution set $S = \{(0, 0)\}$ with the quadratic constant $\kappa = 0.3$. In Figure 2a, the yellow area indicates the part where only the linear component of $f(y)$, i.e. $b^\mathsf{T} y$, is active while the nonlinear penalty component $\rho \max\{0, \lambda_{\max}(\mathcal{A}^*(y) - C)\}$ is 0, and the blue region is the part where both the linear and nonlinear components of $f(y)$ are active. The black line in Figure 2a is the boundary where $\lambda_{\max}(\mathcal{A}^*(y) - C) = 0$ which can be characterized by the nonlinear equation $y_1 y_2 - y_1 - y_2 = 0$ with $y_1 < 1$. This nonlinear direction also indicates that sharp growth of $f$ is impossible. Indeed, let $y_1 \in [0, 1)$ and $y_2 = \frac{y_1}{y_1 - 1}$. Note that this choice of $y$ satisfies the nonlinear equation $y_1 y_2 - y_1 - y_2 = 0$ and thus only the linear part $-b^\mathsf{T} y$ in $f(y)$ is in force. We then have*

$$
f(y) - f^\star = -y_1 - y_2 = -y_1 - \frac{y_1}{y_1 - 1} = \frac{-y_1^2}{y_1 - 1}.
$$

*On the other hand, the distance to the solution set can be verified as*

$$
\mathrm{dist}(y, S) = \sqrt{y_1^2 + \frac{y_1^2}{(y_1 - 1)^2}} = \left| \frac{y_1 \sqrt{1 + (y_1 - 1)^2}}{y_1 - 1} \right| \geq \left| \frac{y_1}{y_1 - 1} \right|.
$$

*It becomes clear that it is impossible to have a constant $\kappa > 0$ such that*

$$
\kappa \left| \frac{y_1}{y_1 - 1} \right| \leq \frac{-y_1^2}{y_1 - 1} = f(y) - f^\star, \quad \forall y_1 \in [0, 1)
$$

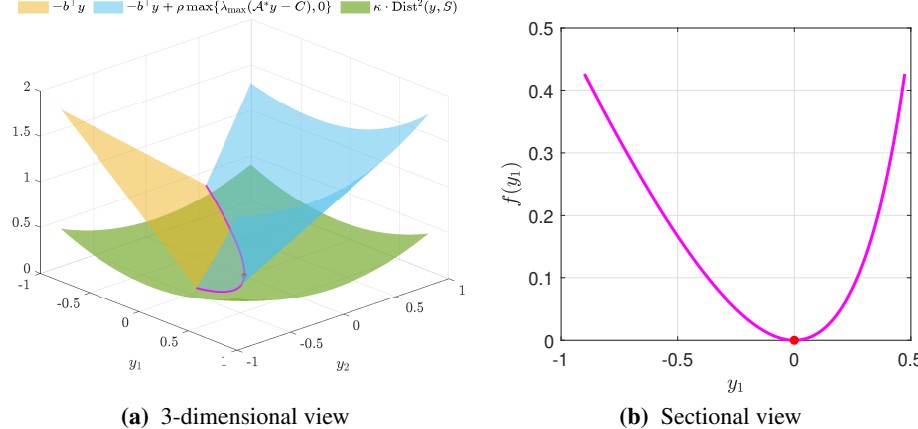

**(a)** 3-dimensional view          **(b)** Sectional view

**Figure 2:** The quadratric growth property of the exact penalty function $f(y) = -b^\mathsf{T}y + \rho \max\{0, \lambda_{\max}(\mathcal{A}^*(y) - C)\}$ where $\rho = 4$ and $f^\star = 0$. The optimal solution set $S = \{(0,0)\}$. In Figure 2a, the yellow region represents the linear part where only $-b^\mathsf{T}y$ is active, resulting $\rho \max\{0, \lambda_{\max}(\mathcal{A}^*(y) - C)\} = 0$, the blue region encompasses both the linear and the nonlinear parts, and the green surf is the square of the distance to the optimal solution set with $\kappa = 0.3$. Figure 2b shows the sectional view of $f$ along the direction $y_1 y_2 - y_1 - y_2 = 0$, which can be characterized as the rational function $f(y_1) = \frac{-y_1^2}{y_1 - 1}$.

as the quadratic therm $y_1^2$ will dominant when $y_1$ is small, let alone the sharp growth $\kappa \cdot \operatorname{dist}(y, S) \leq f(y) - f^\star$. Moreover, the quadratic growth of indicator function formulation $-b^\mathsf{T}y + \delta_{\mathbb{S}_+^2}(C - \mathcal{A}^*(y))$ has also been verified as we only need to consider the linear component $-b^\mathsf{T}y$, which has already been confirmed in the previous case. Lastly, it is worth pointing out that the sublevel sets of the functions $f$ and $-b^\mathsf{T}y + \delta_{\mathbb{S}_+^2}(C - \mathcal{A}^*(y))$ are different. Indeed, given a finite value $\beta > 0$, the sublevel set $\{y \in \mathbb{R}^2 \mid f(y) \leq \beta\}$ contains both parts of yellow and blue regions in Figure 2b. In contrast, the sublevel set $\{y \in \mathbb{R}^2 \mid -b^\mathsf{T}y + \delta_{\mathbb{S}_+^2}(C - \mathcal{A}^*(y)) \leq \beta\}$ only contains the linear part in Figure 2b.

## E  Proofs and supplemental discussions in Section 4

In this section, we complete the missing proof in Section 4 and discuss the ALM applied to (D) with its corresponding convergence properties. This section is divided into three subsections. Appendix E.1 completes the proof of Proposition 2. Appendix E.2 finishes the proof of Theorem 3. Appendix E.3 discusses the ALM applied to (D).

### E.1  Proof of Proposition 2

The proof of Proposition 2 requires a nice characterization of the ordinal dual function and the augmented dual function defined as

$$g_r(w) = \inf_{X \in \mathbb{S}^n} L_r(X, w) \tag{E.1}$$

where $L_r$ is the augmented Lagrangian function (14), recall that we write $w = (y, Z)$.

**Proposition E.1** ([38, Theorem 3.2]). *Let $r > 0$. For all $w \in \mathbb{R}^m \times \mathbb{S}^n$, it holds that*

$$g_r(w) = \min_{X \in \mathbb{S}^n} L_r(X, w) = \max_{u \in \mathbb{R}^m \times \mathbb{S}^n} g_0(u) - \frac{1}{2r}\|u - w\|^2,$$

*where $g_0$ is the dual function in* (13).

Proposition E.1 not only shows the augmented dual function $g_r$ is $g_0$ with a prior but also shows $g_r$ is continuously differentiable. In comparison, the ordinary dual function $g_0$ can be highly nonsmooth. We are ready to prove Proposition 2.

- Part (a) of Proposition 2 comes directly from the PPM convergence Lemma 1 as Lemma 4 and the stopping criteria (A') naturally imply the dual iterate $w_{k+1}$ satisfies $\|w_{k+1} - \text{prox}_{r_k, -g_0}(w_k)\| \le \epsilon_k$ and $\sum_{k=1}^{\infty} \epsilon_k < \infty$.
- The proof of (b) in Proposition 2 is in fact straightforward. Firstly, by Moreau decomposition [49, Exercise 12.22] and (15c), we have

$$r_k X_{k+1} - Z_k = \Pi_{\mathbb{S}^n_+}(r_k X_{k+1} - Z_k) - \Pi_{\mathbb{S}^n_+}(Z_k - r_k X_{k+1}) = \Pi_{\mathbb{S}^n_+}(r_k X_{k+1} - Z_k) - Z_{k+1}.$$

Diving both sides by $r_k$, we conclude $X_{k+1} + (Z_{k+1} - Z_k)/r_k \in \mathbb{S}^n_+$. Therefore, it follows that

$$\text{dist}(X_{k+1}, \mathbb{S}^n_+) \le \|X_{k+1} - (X_{k+1} + (Z_{k+1} - Z_k)/r_k)\| = \|Z_{k+1} - Z_k\|/r_k.$$

Secondly, $\|\mathcal{A}(X_{k+1}) - b\| = \|y_k - y_{k+1}\|/r_k$ comes directly from the update (15b).

Lastly, note that

$$L_{r_k}(X_{k+1}, w_k) = \langle C, X_{k+1} \rangle + \frac{1}{2r_k}(\|w_{k+1}\|^2 - \|w_k\|^2).$$

On the other hand, we have

$$\min_{X \in \mathbb{S}^n} L_{r_k}(X, w_k) = g_{r_k}(w_k) = g_0(u^\star) - \frac{1}{2r_k}\|u^\star - w_k\|^2 \le g_0(u^\star) \le p^\star,$$

where $u^\star$ is the point that achieves the maximum in Proposition E.1 and the last inequality uses weakly duality. Therefore, we have

$$\langle C, X_{k+1} \rangle - p^\star \le L_{r_k}(X_{k+1}, w_k) - \min_{X \in \mathbb{S}^n} L_{r_k}(X, w_k) - \frac{1}{2r_k}(\|w_{k+1}\|^2 - \|w_k\|^2).$$

Since the convergence of $\{w_k\}$ is guaranteed in (a), the quantities $\text{dist}(X_{k+1}, \mathbb{S}^n_+)$, $\|\mathcal{A}(X_{k+1}) - b\|$, and $\langle C, X_{k+1} \rangle - p^\star$ also convergence to zero.
- Part (c) is a direct consequence of part (b) and the fact that $\Omega_P$ is bounded if and only if the set $\{X \in \mathbb{S}^n \mid \text{dist}(X, \mathbb{S}^n_+) \le \gamma_1, \|\mathcal{A}(X) - b\| \le \gamma_2, \langle C, X \rangle - p^\star \le \gamma_3\}$ is bounded for any $\gamma \in \mathbb{R}^3$ [23, page 110].

## E.2 Proof of Theorem 3

In this subsection, we provide a complete proof of part (a) in Theorem 3. The proof is motivated by [26, Theorem 1] which focuses on dual SDPs only. To facilitate proof, we first introduce a standard result in proximal mapping that upper bounds the step length of the proximal step by the distance to the optimal solution set, which can be found in [42, Proposition 1] by setting $z = x$ and $z' = \Pi_S(x)$. We give a simple proof below.

**Proposition E.2.** *Let $f : \mathbb{R}^n \to \bar{\mathbb{R}}$ be a convex function. Denote $S \subseteq \mathbb{R}^n$ as the set of minimizers of $f$, i.e., $S = \arg\min_{x \in \mathbb{R}^n} f(x)$. Suppose $S \neq 0$. Given a point $x \in \mathbb{R}^n$ and a constant $\alpha > 0$, the proximal mapping holds that*

$$\|\text{prox}_{\alpha, f}(x) - x\| \le \text{dist}(x, S).$$

*Proof.* From the optimality condition of proximal mapping, we have

$$f(\text{prox}_{\alpha, f}(x)) + \frac{\alpha}{2}\|\text{prox}_{\alpha, f}(x) - x\|^2 \le f(\Pi_S(x)) + \frac{\alpha}{2}\|\Pi_S(x) - x\|^2$$

$$\implies \frac{\alpha}{2}\|\text{prox}_{\alpha, f}(x) - x\|^2 \le f(\Pi_S(x)) - f(\text{prox}_{\alpha, f}(x)) + \frac{\alpha}{2}\|\Pi_S(x) - x\|^2$$

$$\implies \|\text{prox}_{\alpha, f}(x) - x\| \le \text{dist}(x, S),$$

which completes the proof. $\square$

Using Proposition E.2 and (B'), we arrive at the following key inequality to show the linear convergence of KKT residuals. Proposition 2. It bounds the step length of the dual update by the current distance to the solution set.

**Proposition E.3.** *[26, Lemma 3] Let $\{X_k, w_k\}$ be the sequence generated by (15) under (B'). Then for all $k \ge 1$ with $\delta_k < 1$, it holds that*

$$\|w_{k+1} - w_k\| \le \frac{1}{1 - \delta_k}\text{dist}(w_k, \Omega_D).$$

*Proof.* From the triangle inequality and the stopping criterion (B′), we have

$$\|w_{k+1} - w_k\| - \|w_k - \mathrm{prox}_{r_k,-g_0}(w_k)\| \leq \|w_{k+1} - \mathrm{prox}_{r_k,-g_0}(w_k)\| \leq \delta_k \|w_{k+1} - w_k\|.$$

Rearranging terms and using Proposition E.2 yields

$$(1 - \delta_k)\|w_{k+1} - w_k\| \leq \|w_k - \mathrm{prox}_{r_k,-g_0}(w_k)\| \leq \mathrm{dist}(w_k, \Omega_\mathrm{D}).$$

This completes the proof. □

We are ready to start the proof of part (a) here. As dual strict complementary holds, the negative of the dual function (13), i.e., $-g_0$, satisfies quadratic growth (9b). Using Lemma 4 and stopping criteria (B′), we have $\|w_{k+1} - \mathrm{prox}_{r_k,-g_0}(w_k)\| \leq \delta_k \|w_{k+1} - w_k\|$. Therefore, the linear reduction on $\mathrm{dist}(w_k, \Omega_\mathrm{D})$ comes directly from the PPM convergence Lemma 1, i.e., there exists a $\hat{k}_1$ such that for all $k \geq \hat{k}_1$, it holds that

$$\mathrm{dist}(w_{k+1}, \Omega_\mathrm{D}) \leq \mu_k \mathrm{dist}(w_k, \Omega_\mathrm{D}), \quad \mu_k = \frac{\theta_k + 2\delta_k}{1 - \delta_k} < 1, \quad \theta_k = \frac{1}{\sqrt{2r_k\mu_\mathrm{q}+1}} < 1.$$

We move to show the bound for residuals $\mathrm{dist}(X_{k+1}, \mathbb{S}_+^n), \|\mathcal{A}(X_{k+1}) - b\|$, and $\langle C, X_{k+1} \rangle - p^\star$. Specifically, when $\delta_k < 1$, it holds that

$$\mathrm{dist}(X_{k+1}, \mathbb{S}_+^n) \leq \frac{\|Z_k - Z_{k+1}\|}{r_k} \leq \frac{\|w_k - w_{k+1}\|}{r_k} \leq \frac{1}{(1 - \delta_k)r_k}\mathrm{dist}(w_k, \Omega_\mathrm{D}),$$

$$\|\mathcal{A}(x_{k+1}) - b\| \leq \frac{\|y_k - y_{k+1}\|}{r_k} \leq \frac{\|w_k - w_{k+1}\|}{r_k} \leq \frac{1}{(1 - \delta_k)r_k}\mathrm{dist}(w_k, \Omega_\mathrm{D}),$$

where we use part (b) in Proposition 2, $\|y_k - y_{k+1}\| \leq \|w_k - w_{k+1}\|$, $\|Z_k - Z_{k+1}\| \leq \|w_k - w_{k+1}\|$, and Proposition E.3. Again, using part (b) in Propositions 2 and E.3, when $\delta_k < 1$, we have

$$\begin{aligned}
\langle C, X_{k+1} \rangle - p^\star &\leq \frac{\delta_k^2\|w_{k+1} - w_k\|^2 + \|w_k\|^2 - \|w_{k+1}\|^2}{2r_k} \\
&= \frac{\delta_k^2\|w_{k+1} - w_k\|^2 + (\|w_k\| + \|w_{k+1}\|)(\|w_k\| - \|w_{k+1}\|)}{2r_k} \\
&\leq \frac{\delta_k^2\|w_{k+1} - w_k\|^2 + (\|w_k\| + \|w_{k+1}\|)(\|w_k - w_{k+1}\|)}{2r_k} \\
&\leq \frac{\delta_k^2\|w_{k+1} - w_k\| + \|w_k\| + \|w_{k+1}\|}{2r_k(1 - \delta_k)}\mathrm{dist}(w_k, \Omega_\mathrm{D}).
\end{aligned}$$

Since $\delta_k \to 0$, there exists a $\hat{k}_2$ such that $\delta_k < 1$ for all $k \geq \hat{k}_2$. Taking $\hat{k} = \max\{\hat{k}_1, \hat{k}_2\}$ completes the proof for part (a) with the constants

$$\begin{aligned}
\mu_k' &= \frac{1}{(1 - \delta_k)r_k} \to \mu_\infty' = \frac{1}{r_\infty}, \\
\mu_k'' &= \frac{\delta_k^2\|w_{k+1} - w_k\| + \|w_k\| + \|w_{k+1}\|}{2r_k(1 - \delta_k)} \to \mu_\infty'' = \frac{\|w_\infty\|}{r_\infty}.
\end{aligned}$$

We move on to part (b). By Theorem 2, we know that the error bound (7b) holds as dual strict complementary holds, i.e., for any bounded set $\mathcal{U} \subseteq \mathbb{S}^n$ containing any $X^\star \in \Omega_\mathrm{P}$, there exist constants $\alpha_1, \alpha_2, \alpha_3 > 0$ such that

$$\langle C, X \rangle - \langle C, X^\star \rangle + \alpha_1\|\mathcal{A}(X) - b\| + \alpha_2 \cdot \mathrm{dist}(X, \mathbb{S}_+^n) \geq \alpha_3 \cdot \mathrm{dist}^2(X, \Omega_\mathrm{P}), \quad \forall X \in \mathcal{U}. \quad \text{(E.2)}$$

From the assumption that $\Omega_\mathrm{P}$ is bounded, Proposition 2 confirms that the sequence $\{X_k\}$ is bounded. Thus, we can choose a bounded set $\mathcal{U} \supseteq \Omega_\mathrm{P} \bigcup \cup_{k \geq 1}\{X_k\}$. Combing (E.2) with (16), we have

$$\alpha_3 \cdot \mathrm{dist}^2(X_{k+1}, \Omega_\mathrm{P}) \leq (\mu_k'' + (\alpha_1 + \alpha_2)\mu_k') \cdot \mathrm{dist}(w_k, \Omega_\mathrm{D}).$$

Dividing both sides by $\alpha_3$ leads to the desired result in part (b). This completes the proof.

### E.3 ALMs for dual conic program (D)

In aiming to offer a comprehensive understanding of ALMs for solving (P) and (D), in this subsection, we provide the algorithm of ALM applied to solving the dual problem (D) along with the convergence analysis. This can be considered as an extension of the work in [26] and a counterpart to the ALM for solving (P) discussed in Section 4.

We first reformulate (D) as the equivalent minimization problem

$$-d^\star = \min_{y\in\mathbb{R}^m, Z\in\mathbb{S}^n} \quad \langle -b, y\rangle$$
$$\text{subject to} \quad C - \mathcal{A}^*(y) \in \mathbb{S}^n_+.$$

We then introduce a dual variable $X \in \mathbb{S}^n_+$ and define the ordinary Lagrangian function as $L_0(y, X) = -\langle b, y\rangle + \langle X, C - \mathcal{A}^*(y)\rangle$. The corresponding dual function and dual problem become

$$g_0(X) = \inf_{y\in\mathbb{R}^m} L_0(y, X) \quad \text{and} \quad \max_{X\in\mathbb{S}^n_+} g_0(X). \tag{E.3}$$

Then, the augmented Lagrangian with parameter $r > 0$, analogous to (14), is

$$L_r(y, X) = \langle -b, y\rangle + \frac{1}{2r}(\|\Pi_{\mathbb{S}^n_+}(X - r(C - \mathcal{A}^*(y)))\|^2 - \|X\|^2).$$

Given an initial point $X_0 \in \mathbb{S}^n$ and a sequence of positive scalars $r_k \uparrow r_\infty$, the ALM generates a sequence of $\{y_k\}$ and $\{X_k\}$ following

$$y_{k+1} \approx \operatorname*{argmin}_{y\in\mathbb{R}^m} L_{r_k}(y, X_k) \tag{E.4a}$$

$$X_{k+1} = X_k + r_k \nabla_X L_{r_k}(y_{k+1}, X_k) = \Pi_{\mathbb{S}^n_+}(X_k - r_k(C - \mathcal{A}^*(y_{k+1}))), \quad k = 0, 1, \ldots. \tag{E.4b}$$

Many of the same properties discussed in Section 4 hold. We will provide the precise statement in the remainder of this section. Similarly, we can consider the analogous stopping criteria of (A′) and (B′).

$$L_{r_k}(y_{k+1}, X_k) - \min_{y\in\mathbb{R}^m} L_{r_k}(y, X_k) \le \epsilon_k^2/(2r_k), \quad \textstyle\sum_{k=1}^\infty \epsilon_k < \infty, \tag{A′_D}$$

$$L_{r_k}(y_{k+1}, X_{k+1}) - \min_{y\in\mathbb{R}^m} L_{r_k}(y, X_k) \le \delta_k^2\|X_{k+1} - X_k\|^2/(2r_k), \quad \textstyle\sum_{k=1}^\infty \delta_k < \infty. \tag{B′_D}$$

**Proposition E.4.** *Consider* (P) *and* (D). *Assume strong duality holds and $\Omega_\mathrm{P} \ne \emptyset$. Let $\{y_k, X_k\}$ be a sequence from the ALM* (E.4) *under* (A′_D) *and write $Z_k = C - \mathcal{A}^*(y_k)$ for all $k \ge 0$. The following statements hold.*

(a) *The dual sequence $X_k$ is bounded. Further, $\lim_{k\to\infty} X_k = X_\infty \in \Omega_\mathrm{P}$ (i.e., the whole sequence converges to a primal optimal solution).*

(b) *The dual feasibility and cost value gap satisfy*

$$\operatorname{dist}(Z_{k+1}, \mathbb{S}^n_+) \le \|X_k - X_{k+1}\|/r_k \to 0,$$
$$d^\star - \langle b, y_{k+1}\rangle \le L_{r_k}(y_{k+1}, X_k) - \min_{y\in\mathbb{R}^m} L_{r_k}(y, X_k) + (\|X_k\|^2 - \|X_{k+1}\|^2)/(2r_k) \to 0.$$

(c) *If the solution set $\Omega_\mathrm{D}$ in* (1b) *is nonempty and bounded, then the primal sequence $y_k$ is also bounded, and all of its cluster points belongs to $\Omega_\mathrm{D}$.*

*Proof.* The proof of many of these statements is a change of the notations in the proof in Proposition 2.

- Part (a) comes directly from the PPM convergence by noting Lemma 4 and the stopping criterion (A′_D) imply the dual iterate $X_{k+1}$ satisfies $\|X_{k+1} - \operatorname{prox}_{r_k, -g_0}(X_k)\| \le \epsilon_k$.
- By the Moreau decomposition [49, Exercise 12.22], we have

$$r_k Z_{k+1} - X_k = \Pi_{\mathbb{S}^n_+}(r_k Z_{k+1} - X_k) - \Pi_{\mathbb{S}^n_+}(X_k - r_k Z_{k+1}) = \Pi_{\mathbb{S}^n_+}(r_k X_{k+1} - Z_k) - X_{k+1}.$$

Thus, $Z_{k+1} + \frac{X_{k+1} - X_k}{r_k} \in \mathbb{S}^n_+$. Therefore,

$$\operatorname{dist}(Z_{k+1}, \mathbb{S}^n_+) \le \left\| Z_{k+1} - \left( Z_{k+1} + \frac{X_{k+1} - X_k}{r_k}\right)\right\| = \frac{1}{r_k}\|X_{k+1} - X_k\|.$$

From the definition of the augmented Lagrangian function, we have

$$L_{r_k}(y_{k+1}, X_k) = \langle -b, y_{k+1} \rangle + \frac{1}{2r}(\|X_{k+1}\|^2 - \|X_k\|^2).$$

On the other hand, we know

$$\min_{y \in \mathbb{R}^m} L_{r_k}(y, X_k) = g_0(U^\star) - \frac{1}{2r_k}\|U^\star - X_k\|^2 \leq g_0(U^\star) \leq -d^\star,$$

where $U^\star$ denoted as the point that achieves the maximum in the identity [38, Theorem 3.2]

$$\min_{y \in \mathbb{R}^m} L_{r_k}(y, \cdot) = \max_{X \in \mathbb{S}^n} g_0(X) - \frac{1}{2r_k}\|X - \cdot\|^2$$

and the last inequality comes from the definition of the dual problem in (E.3). Thus,

$$d^\star - \langle b, y_{k+1} \rangle \leq L_{r_k}(y_{k+1}, X_k) - \min_{y \in \mathbb{R}^m} L_{r_k}(y, X_k) - \frac{1}{2r}(\|X_{k+1}\|^2 - \|X_k\|^2).$$

- Part (c) is a consequence of part (b) and the fact that $\Omega_D$ is bounded if and only if the set $\{(y, Z) \in \mathbb{R}^m \times \mathbb{S}^n \mid d^\star - \langle b, y \rangle \leq \gamma_1, \mathrm{dist}(Z, \mathbb{S}^n_+) \leq \gamma_2, \|C - \mathcal{A}^*(y) - Z\| \leq \gamma_3\}$ is bounded for any $\gamma \in \mathbb{R}^3$ [23, page 110]. $\qquad\square$

**Theorem E.1** (Linear convergences). *Consider primal and dual SDPs* (P) *and* (D). *Assume strong duality and strict complementarity holds (implying $\Omega_P \neq \emptyset$ and $\Omega_D \neq \emptyset$). Let $\{y_k, X_k\}$ be a sequence from the ALM* (E.4) *under* $(A'_D)$ *and* $(B'_D)$ *and write $Z_k = C - \mathcal{A}^*(y_k)$ for all $k \geq 0$. The following statements hold.*

(a) *There exists constants $\hat{k} \geq 0$, $0 < \mu_k < 1$ and $\mu'_k, \mu''_k > 0$ such that for all $k \geq \hat{k}$,*

$$\mathrm{dist}(X_{k+1}, \Omega_P) \leq \mu_k \cdot \mathrm{dist}(X_k, \Omega_P),$$
$$\mathrm{dist}(Z_{k+1}, \mathbb{S}^n_+) \leq \mu'_k \cdot \mathrm{dist}(X_k, \Omega_P), \tag{E.5}$$
$$d^\star - \langle b, y_{k+1} \rangle \leq \mu''_k \cdot \mathrm{dist}(X_k, \Omega_P).$$

(b) *If $\Omega_D$ is bounded, then the primal sequence $\{y_k\}$ also converges linearly to $\Omega_D$, i.e., there is a constant $\hat{k} \geq 0$ such that for all $k \geq \hat{k}$,*

$$\mathrm{dist}^2(y_{k+1}, \Omega_P) \leq \tau_k \cdot \mathrm{dist}(X_k, \Omega_D).$$

*Proof.* Similar to Proposition E.3, if $(B'_D)$ is used, we have

$$\|X_{k+1} - X_k\| \leq \frac{1}{1 - \delta_k}\mathrm{dist}(X_k, \Omega_P), \quad \forall \delta_k < 1. \tag{E.6}$$

Therefore, there exists a $\hat{k} \geq 0$ such that $\forall k \geq \hat{k}, \delta_k < 1$ and

$$\mathrm{dist}(Z_{k+1}, \mathbb{S}^n_+) \leq \frac{\|X_{k+1} - X_k\|}{r_k} \leq \frac{1}{r_k(1 - \delta_k)}\mathrm{dist}(X_{k+1}, \Omega_P),$$

where the first inequality uses part (b). Then, once again using (E.6) and part (b), we have

$$\begin{aligned}
d^\star - \langle b, y \rangle &\leq \frac{\delta_k^2 \|X_{k+1} - X_k\|^2 + \|X_k\|^2 - \|X_{k+1}\|^2}{2r_k} \\
&\leq \frac{\delta_k^2 \|X_{k+1} - X_k\|^2 + (\|X_k\| - \|X_{k+1}\|)(\|X_k\| + \|X_{k+1}\|)}{2r_k} \\
&\leq \frac{\delta_k^2 \|X_{k+1} - X_k\|^2 + (\|X_k - X_{k+1}\|)(\|X_k\| + \|X_{k+1}\|)}{2r_k} \\
&\leq \frac{\delta_k^2 \|X_{k+1} - X_k\| + \|X_k\| + \|X_{k+1}\|}{2r_k(1 - \delta_k)}\mathrm{dist}(X_{k+1}, \Omega_P).
\end{aligned}$$

If strict complementarity holds, Theorem 1 (with $g$ in (5) as an indicator function) guarantees that $-g_0$ in (E.3) satisfies quadratic growth in (9b). Thus, the linear convergence of $\mathrm{dist}(X_k, \Omega_P)$ comes from the PPM convergence Lemma 1, i.e., there exists a $\hat{k}_1$ such that for all $k \geq \hat{k}_1$, it holds that

$$\mathrm{dist}(X_{k+1}, \Omega_P) \leq \mu_k \mathrm{dist}(X_k, \Omega_P), \quad \mu_k = \frac{\theta_k + 2\delta_k}{1 - \delta_k} < 1, \quad \theta_k = \frac{1}{\sqrt{2r_k\mu_q + 1}} < 1.$$

Since $\delta_k \to 0$, there exists a $\hat{k}_2$ such that $\delta_k < 1$ for all $k \geq \hat{k}_2$. Taking $\hat{k} = \max\{\hat{k}_1, \hat{k}_2\}$ completes the proof for part (a) with the constants

$$\mu'_k = \frac{1}{r_k(1 - \delta_k)} \to \mu'_\infty = \frac{1}{r_\infty},$$

$$\mu''_k = \frac{\delta_k^2 \|X_{k+1} - X_k\| + \|X_k\| + \|X_{k+1}\|}{2r_k(1 - \delta_k)} \to \frac{\|X_\infty\|}{r_\infty}.$$

On the other hand, Theorem 1 also guarantees that the error bound (8b) holds, i.e., for any bounded set $\mathcal{V} \supseteq \Omega_{\mathrm{D}}$, there exist constants $\alpha_1, \alpha_2, \alpha_3 > 0$ such that

$$d^\star - \langle b, y \rangle + \alpha_1 \|C - \mathcal{A}^*(y) - Z\| + \alpha_2 \cdot \mathrm{dist}(Z, \mathbb{S}_+^n) \geq \alpha_3 \cdot \mathrm{dist}^2((y, Z), \Omega_{\mathrm{D}}), \quad \forall (y, Z) \in \mathcal{V}. \quad \text{(E.7)}$$

By the design of (E.4), we always have $\|C - \mathcal{A}^*(y_k) - Z_k\| = 0$. By Proposition E.4 - part (c), the sequence $\{y_k, Z_k\}$ is bounded. Thus, choosing a bounded set $\mathcal{V} \supseteq \Omega_{\mathrm{D}} \bigcup \cup_{k \geq 1} \{y_k, Z_k\}$ and combing (E.7) with (E.5), we have

$$\alpha_3 \cdot \mathrm{dist}^2(y_{k+1}, \Omega_{\mathrm{D}}) \leq (\mu''_k + \alpha_2 \mu'_k) \cdot \mathrm{dist}(X_k, \Omega_{\mathrm{P}}), \ \forall k \geq \hat{k}.$$

Dividing both sides by $\alpha_3$ leads to the desired result. This completes the proof. $\qquad\square$

## F    Details of the numerical experiments

In Section 5, we consider the classical SDP relaxation of the Mac-Cut problem and the linear SVM. In this section, along with another popular signal processing problem - Lasso, we detail the problem formulation and report the numerical performance of ALM applied to those problems.

- The SDP relaxation of Max-Cut:

$$\min_{X \in \mathbb{S}^n} \quad \langle C, X \rangle$$
$$\text{subject to} \quad \mathrm{Diag}(X) = 1,$$
$$X \in \mathbb{S}_+^n,$$

where $C \in \mathbb{S}^n$ is the problem data representing a weighted undirected graph, $1 \in \mathbb{R}^n$ is an all one vector, and $\mathrm{Diag}(X) = [X_{11}, \ \cdots \ , X_{nn}]^{\mathsf{T}}$. This type of problem has been shown to satisfy strict complementarity and has unique primal and dual solutions for almost all data matrix $C$ [41, Corollary 2.3].
- Linear SVM:

$$\min_{x \in \mathbb{R}^d} \lambda \sum_{i=1}^m \max\{0, 1 + b_i(a_i^{\mathsf{T}} x)\} + \frac{1}{2}\|x\|^2,$$

where $a_i \in \mathbb{R}^n$ is the feature of the $i$-th data point, $b_i \in \{-1, 1\}$ is the corresponding label, and $\lambda > 0$ is a constant. Equivalently, the problem can reformulated in the following standard conic form

$$\min_{x \in \mathbb{R}^n, t \in \mathbb{R}^m} \quad \frac{1}{2}\|x\|^2 + \lambda \mathbf{1}^{\mathsf{T}} t$$
$$\text{subject to} \quad \mathrm{Diag}(b)Ax + \mathbf{1} \leq t,$$
$$0 \leq t,$$

where $A = [a_1, \cdots, a_m]^{\mathsf{T}} \in \mathbb{R}^{m \times n}$ and $\mathrm{Diag}(b) \in \mathbb{S}^m$ denotes the diagonal matrix with the diagonal elements being $b$.
- Lasso:

$$\min_{x \in \mathbb{R}^n} \quad \frac{1}{2}\|Ax - b\|^2 + \lambda \|x\|_1,$$

where $A \in \mathbb{R}^{m \times n}, b \in \mathbb{R}^m$ are problem data and $\lambda > 0$ is the weighted parameter balancing the sparsity of the solution and the accuracy of the linear solution. Equivalently, the problem can be rewritten in the conic form

$$\min_{x \in \mathbb{R}^n} \quad \frac{1}{2}\|y\|^2 + \lambda \mathbf{1}^{\mathsf{T}} t$$
$$\text{subject to} \quad Ax - b = y,$$
$$-t \leq x \leq t.$$

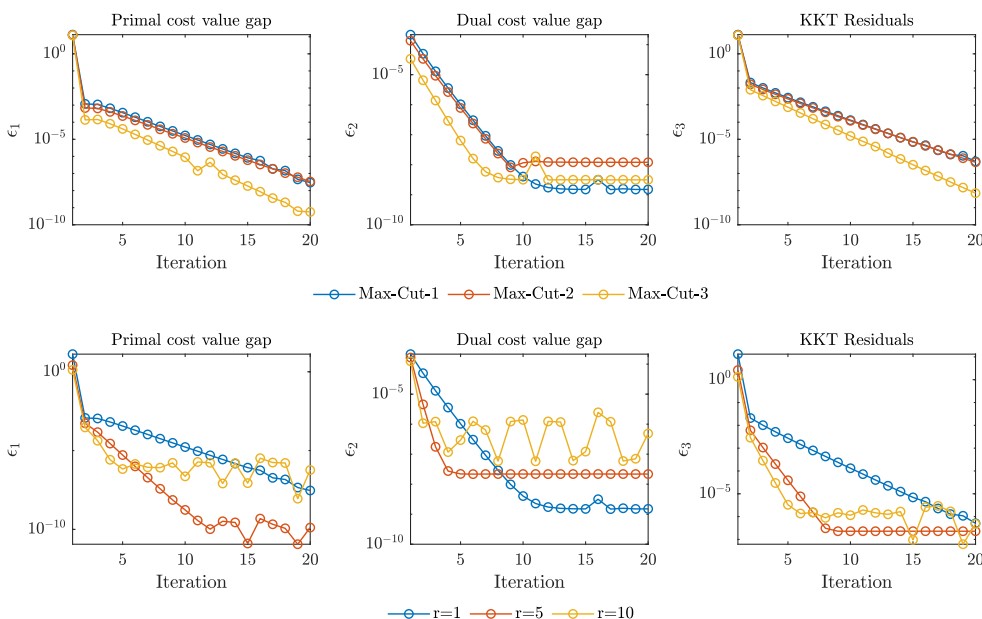

**Figure 3:** Nuremical experiments for Max-Cut with different instances and various augmented Lagrangian parameter $r > 0$.

All numerical experiments are conducted on a PC with a 12-core Intel i7-12700K CPU@3.61GHz and 32GB RAM. For Max-Cut problem, we select the graph $G_1, G_2$, and $G_3$ from the website `https://web.stanford.edu/~yyye/yyye/Gset/` and only take the first $20 \times 20$ submatrix as the considered problem data $C$. For both the linear SVM and Lasso, we randomly generate the problem data with the dimension $m = 100$ and $n = 10$, and set the constant $\lambda = 1$. We use Mosek [56] to get the optimal cost value. In each application, we consider the following two scenarios 1) fixing the same augmented parameter $r_k = r$ for all $k$ for three different problem instances; 2) fixing a problem instance and varying different augmented parameter $r$. The numerical results are presented in Figures 3 to 5. For each Figures 3 to 5, the first row shows the evolution of the primal cost gap, dual cost gap, and KKT residuals for three different instances with a fixed augmented parameter (scenario one), and the second row compares the different numerical behaviors when varying the augmented parameter (scenario two). In the setting of a fixed augmented parameter, the augmented term $r$ is set as $1, 5, 20$ for Max-Cut, linear SVM, and Lasso respectively. In the setting of various augmented parameters, the parameter is chosen as $r = \{1, 5, 10\}$, $r = \{1, 5, 10\}$, and $r = \{10, 20, 30\}$ for Max-Cut, linear SVM, and Lasso respectively.

We observe that, in all cases, ALM enjoys linear convergence in the primal cost value gap, dual cost value gap, and KKT residuals to the accuracy of at least $10^{-5}$, which is consistent with our theoretical findings in Theorem 3. We believe the oscillated or flattening behavior that appears in the tail (when the iterates are close to the solution set) is due to the inaccuracy of the subproblem solver and computational errors. The numerical result also indicates that when the augmented term $r$ increases, ALM favors the decrease of the feasibility more, leading to faster convergence in the beginning phase. However, a large $r$ may not lead to the best convergence in the long term.

Note that the projection term $\|\Pi_{\mathbb{S}^n_+}(Z - rX)\|^2$ in the subproblem (15a) can not be directly modeled by Yalmip. Therefore, we reformulate the problem as the following

$$\underset{X \in \mathbb{S}^n}{\operatorname{argmin}} \langle C, X \rangle + \frac{1}{2r}(\|y + r_k(b - \mathcal{A}(X))\|^2 + \|\Pi_{\mathbb{S}^n_+}(Z - r_k X)\|^2 - \|y\|^2 - \|Z\|^2)$$

$$= \underset{X \in \mathbb{S}^n}{\operatorname{argmin}} \langle C, X \rangle + \frac{1}{2r}(\|y + r_k(b - \mathcal{A}(X))\|^2 + \|\Pi_{\mathbb{S}^n_-}(r_k X - Z)\|^2)$$

$$= \underset{X \in \mathbb{S}^n, U \in \mathbb{S}^n_+}{\operatorname{argmin}} \langle C, X \rangle + \frac{1}{2r}(\|y + r_k(b - \mathcal{A}(X))\|^2 + \|r_k X - Z - U\|^2),$$

where the first equality drops two independent terms and uses $-\mathbb{S}_+^n = \mathbb{S}_-^n$ and $\|\Pi_{\mathbb{S}_+^n}(Y)\| = \| -\Pi_{(-\mathbb{S}_+^n)}(-Y)\| = \|\Pi_{(\mathbb{S}_-^n)}(-Y)\|$ for all $Y \in \mathbb{S}^n$, and the last equality uses $\min_{U \in \mathbb{S}_+^n} \|Y - U\| = \|Y - \Pi_{\mathbb{S}_+^n}(Y)\| = \|\Pi_{\mathbb{S}_-^n}(Y)\|$ for all $Y \in \mathbb{S}^n$.

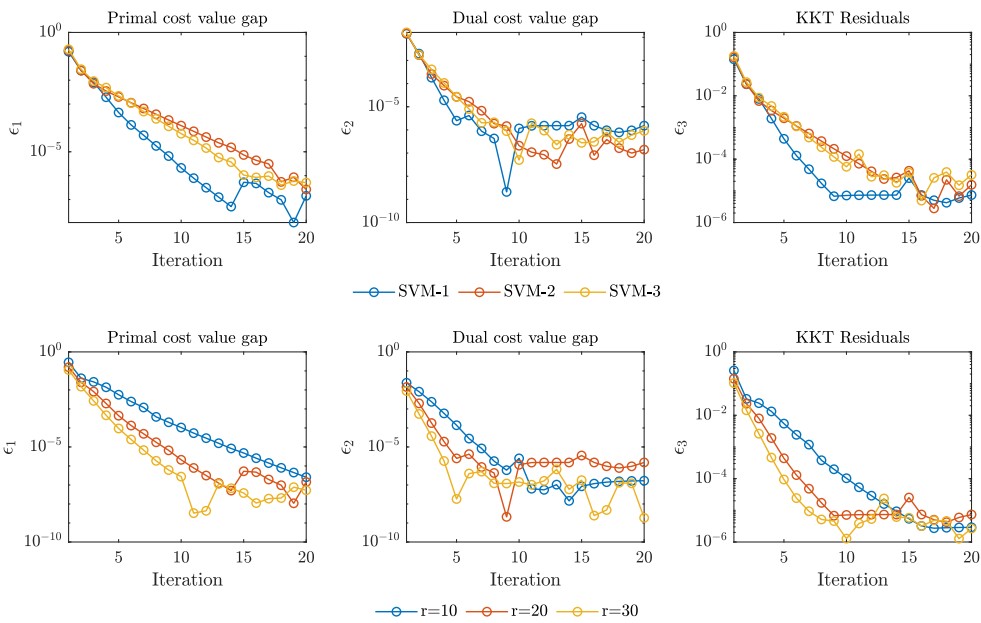

**Figure 4:** Nuremical experiments for linear SVM with different instances and various augmented Lagrangian parameter $r > 0$.

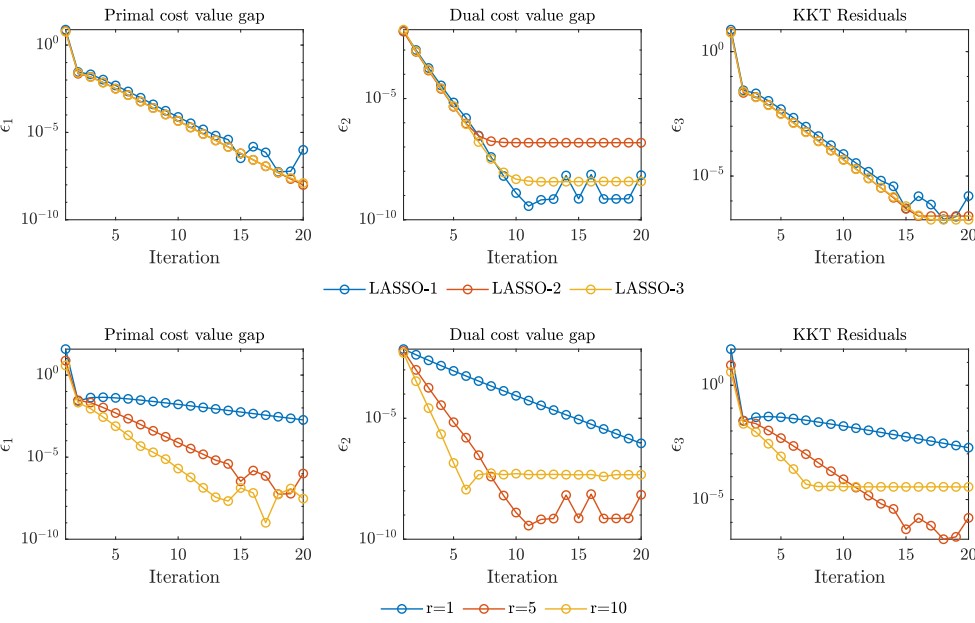

**Figure 5:** Nuremical experiments for Lasso with different instances and various augmented Lagrangian parameter $r > 0$.

