# OpenReview forum: "Inexact Augmented Lagrangian Methods for Conic Optimization: Quadratic Growth and Linear Convergence"
_NeurIPS.cc/2024/Conference — NeurIPS 2024 poster_

### Official Review · Reviewer_rMRz · 2024-07-08

**Soundness:** 3
**Presentation:** 2
**Contribution:** 2
**Rating:** 5
**Confidence:** 3

**Summary:**

This paper presents new theoretical results on the convergence of primal iterates in inexact augmented Lagrangian methods (ALMs) for conic optimization. The idea is to use strict complementarity (which is a standard assumption in conic optimization) to establish quadratic growth and error bound conditions for the primal and dual programs. These bounds are then used to show linear convergence for both the primal and dual iterates in inexact ALMs. The linear convergence of the primal iterates is shown in (brief) experiments.

**Strengths:**

* The work provides a thorough introduction to the problem background
* The theory is presented rigorously!

**Weaknesses:**

I found the main weakness of the paper to be in its presentation of results.

* The paper takes too long to reach the main results. Perhaps an informal version of Theorem 3 could be stated in the introduction?

* There is a lot of notation/specific terminology in the paper, which makes it difficult to follow. One suggestion to reduce the complexity would be to focus on a particular problem class (LP, SOCP, or SDP) in the main paper. For example, if focusing on LPs, the authors could move the discussion of strict complementarity of LPs to earlier in the paper, which would give more intuition for the assumptions in the paper. Furthermore, the theorems in the main paper could be simplified to address LPs, which would make the results easier to understand for a wider audience. The general results for LPs, SOCPs, and SDPs could then be presented in the appendix.

* I would have liked the paper to expound upon how their work relates to machine learning (outside of some brief citations in the introduction). Ideally, this could be done by showing linear convergence of primal iterates on a conic program from machine learning. One idea could be to solve an SVM problem (QP, which is a special case of SDP).

* While Figure 1 suggests that primal iterates are converging linearly, I would like to see more iterations (say 20) in the plots to really verify this claim.

* I found it hard to follow how the author’s results compare to previous convergence results in the literature. I’d suggest providing a more in-depth comparison in lines 341-349.

* The paper takes too long to discuss inexact augmented Lagrangian methods (these are not discussed in-depth until page 7). This makes the flow of the paper confusing — I’d suggest covering inexact ALMs closer to the introduction.

Minor comments:

* Discussion of scalable first-order methods (line 28) should cite PDLP [1, 2]

* “Lipschitz” is misspelled on lines 100 and 103

[1] Applegate, David, Mateo Diaz, Oliver Hinder, Haihao Lu, Miles Lubin, Brendan ODonoghue, and Warren Schudy. 2021. “Practical Large-Scale Linear Programming Using Primal-Dual Hybrid Gradient.” In Advances in Neural Information Processing Systems, edited by M. Ranzato, A. Beygelzimer, Y. Dauphin, P. S. Liang, and J. Wortman Vaughan, 34:20243–57. Curran Associates, Inc.

[2] Applegate, D., Hinder, O., Lu, H. et al. Faster first-order primal-dual methods for linear programming using restarts and sharpness. Math. Program. 201, 133–184 (2023).

**Questions:**

* Are there problems for which the dual iterates can converge linearly, but the primal iterates converge sublinearly? If so, could this be demonstrated in an experiment?

* Line 265: What exactly is meant by “nice self-dual structure”?

**Limitations:**

Please see “Weaknesses”

---

> ### Author Rebuttal · Authors · 2024-08-06
>
> The authors are very thankful to the reviewer for taking the time and effort to review our manuscript. We sincerely appreciate all your valuable comments. All your comments are carefully addressed below.
>
> > **Suggestion focusing on a specific cone, for example, LP. This will make the paper easier to read**
>
> Thanks for the suggestion. This will definitely make the paper more accessible. In this paper, we aim to make the paper as general as possible. In addition, there are some differences in terms of the construction of the penalty function and region/power term for the error bound. In our current manuscript, we have incorporated LP, SOCP, and SDP in our main text. Detailed proofs and individual discussions for each case are provided in the Appendix.
>
> However, we do agree with the reviewer the it will improve the paper clarity by focusing on one problem class. If the paper is accepted, we will follow the reviewer' suggestions and focus on the general problem class—SDP—in the main manuscript. This will not only improve clarity but also give us more space to ensure that Theorem 3 is well-integrated into the main discussion.
>
> > **Expound upon how their work relates to machine learning.**
>
> We have added more numerical experiments in the attached PDF file under the section of Auther rebuttal. We consider the application of linear SVM and Lasso. Both the linear SVM and Lasso can be reformulated as standard conic programs with free, nonnegative, and second-order cone variables. For each application, we randomly generate three instances and run ALM for those three instances with various augmented penalty parameters $\rho>0$. The numerical results show that the three quantities (primal cost value gap, dual cost value gap, and the KKT residuals) all converge to zero with a linear rate. The convergence speed of the residuals also depends on the value of the penalty parameter $\rho$. The larger the penalty parameter $\rho$, the faster the convergence of the residuals. We believe the oscillated or flattening behavior that happens in the tail (when the iterates are close to the solution set) is due to the inaccuracy of the subproblem solver and computational errors.
>
> > **While Figure 1 suggests that primal iterates are converging linearly, I would like to see more iterations (say 20) in the plots to really verify this claim.**
>
> Thanks for pointing this out. However, as we use the modeling package YALMIP to formulate the subproblem and call the conic solver MOSEK to solve the subproblem. It will be difficult to control the subproblem solution accuracy. There exists oscillated and flattening behavior when the iterates are close to the solution set due to the computational error. This is reflected in our additional numerical experiments.
>
> > **I found it hard to follow how the author’s results compare to previous convergence results in the literature. I’d suggest providing a more in-depth comparison in lines 341-349.**
>
> Thanks for the comments. We will revise the comparison in lines 341-349. Specifically, we mean that
> * [24, Theorem 5] requires two things:
>   1. The Lagrangian function is Lipschitz continuous at the origin, which requires both the primal and the dual solution to be unique. This assumption can easily fail, as pointed out in  [27, Section 344 3].
>   2. One more subproblem stopping criterion in addition to $A^{'}$ and $B^{’}$
>
>  Our result in Theorem 3, however, suggests that the linear convergence of the primal iterates also can happen under the standard assumption of strict complementarity and bounded primal solution set.
>
> * As strict complementarity is a generic property of semidefinite programs  [42, Theorem 15] and a bounded primal solution set happens when a dual slater point exists, which is a common assumption, therefore, our result suggests that the linear convergence of the primal iterates is likely to happen under many nicely behaved problems.
>
> [24] R Tyrrell Rockafellar. Augmented Lagrangians and applications of the proximal point algorithm in convex programming. Mathematics of operations research, 1(2):97–116, 1976.
>
>  [27] Ying Cui, Defeng Sun, and Kim-Chuan Toh. On the R-superlinear convergence of the KKT residuals generated by the augmented Lagrangian method for convex composite conic programming. Mathematical Programming, 178:381–415, 2019.
>
> [42] Farid Alizadeh, Jean-Pierre A Haeberly, and Michael L Overton. Complementarity and nondegeneracy in semidefinite programming. Mathematical programming, 77(1):111–128, 1997.
>
> > **The paper takes too long to discuss inexact augmented Lagrangian methods (these are not discussed in-depth until page 7). This makes the flow of the paper confusing — I’d suggest covering inexact ALMs closer to the introduction.**
>
> Thanks for your nice suggestion. We will revise the introduction and incorporate further discussions about Theorem 3.
>
> > **Minor comments:**
> > 1. Discussion of scalable first-order methods (line 28) should cite PDLP [1, 2]
> > 2. “Lipschitz” is misspelled on lines 100 and 103
>
> Thanks for the detailed examination of our paper. We will add the reference in the final version and fix the typo.
>
> > **Are there problems for which the dual iterates can converge linearly, but the primal iterates converge sublinearly? If so, could this be demonstrated in an experiment?**
>
> Thanks for this insightful question. We believe it is a difficult task as strict complementarity holds for almost all conic problems; thus, for many instances, both primal and dual iterates are expected to converge linearly. Right now, we are struggling to find an explicit example for which the dual iterates can converge linearly, but the primal iterates converge sublinearly.
>
> > **Line 265: What exactly is meant by “nice self-dual structure”?**
>
> We simply mean the cones of nonnegative orthant, second-order cone, and the positive semidefinite cones are self-dual. As a result, both the primal and the dual problem are the same class of problems.

---

> > ### Comment · Reviewer_rMRz · 2024-08-11
> > **Thanks for the rebuttal**
> >
> > Thank you for taking the time to address my concerns. I will raise my score from 3 -> 5.

---

### Official Review · Reviewer_X2mJ · 2024-07-11

**Soundness:** 3
**Presentation:** 3
**Contribution:** 4
**Rating:** 8
**Confidence:** 3

**Summary:**

The paper presents the convergence rate of the primal iterates of the augmented Lagrangian Methods (ALMs) which are widely employed in solving constrained optimizations. The authors develop new quadratic growth and error bound properties for primal and dual conic programs under the standard strict complementarity condition and then reveal that both primal and dual iterates of the ALMs converge linearly contingent upon the assumption of strict complementarity and a bounded solution set. The specific contributions include:
1. Under the standard strict complementarity assumption, the quadratic growth and error bound for both primal and dual conic programs (P) and (D) over any compact set containing an optimal solution (see Theorems 1 and 2) are established.
2. A new characterization of the preimage of the subdifferential of the exact penalty functions is unveiled.
3. They provide new and simple proof for the growth properties in the exact penalty functions and clarify some subtle differences in constructing exact penalty functions.
4. They show the linear convergence for both the primal and dual iterates of  ALM to solve either the primal or dual conic programs.

**Strengths:**

Originality: The paper introduces novel quadratic growth and error bound properties for primal and dual conic programs, addressing long-standing challenges in the field. The development of symmetric inexact ALMs for primal and dual problems is a creative extension of existing methods.

Quality: The paper is of high quality, featuring a robust theoretical framework and rigorous mathematical proofs.

Clarity: The paper is clearly written and well-organized.

Significance: By addressing open questions and providing new theoretical insights, the work could influence future research.

**Weaknesses:**

1. The authors claim in Remark 1 that the results in Theorems 1 and 2 are more general and unified compared with known results since they allow for any compact set $\mathcal{U}$. However, let $\mathcal{U}=\mathcal{U}_1\cap \mathcal{U}_2$ where $\mathcal{U}_1$ and $\mathcal{U}_2$  are compact, and there is not any optimal solution in $\mathcal{U}_1$. If it is known that the error bound is constrained with a pair $\gamma, \kappa$ for $x\in\mathcal{U}_2$. For $x\in\mathcal{U}_1$, it is natural to get a $\kappa$ through the minimizatin of the fraction $$\frac{f(x)-p^*+\gamma\|\mathcal{A}(x)-b\|}{dist^p(x,\Omega)}$$ since $\mathcal{U}_1$ is compact and $dist^p(x,\Omega)>0$. We can just take the minimizer of two $\kappa$(s). The advantages of this error bound form in this paper is doubtable.
2. Aftrer Definition 2, the authors claim that the notion of strict complementarity is not restrictive, and it holds generically for conic programs. After that,  the authors also presented: that it has been revealed that many structured SDPs from practical applications also satisfy strict complementarity. There are contradictions and misunderstandings in these two assertions.

**Questions:**

1. Lines 184-185: If the linear constraints $\mathcal{A}(x)=b$ is empty, how to define the exact penalty function?
2. Line 226: Why is there $\mathbb{F}\times \mathbb{E}$ instead of $\mathbb{F}$?
3. Please consider citing the paper "Rockafellar, R.T. Convergence of Augmented Lagrangian Methods in Extensions Beyond Nonlinear Programming. Math. Program. 199, 375–420 (2023). https://doi.org/10.1007/s10107-022-01832-5," and compare it with your work.

Typos:
1. Line 238: Should (8b) be replaced by (8a)?
2. The notation $\mathbb{R}^n$ should be changed to $\mathbb{R}_{+}^n$.

---

> ### Author Rebuttal · Authors · 2024-08-06
>
> We appreciate the reviewer's time and effort in evaluating our manuscript. Your comments helped further improve the quality of our work. We provide detailed responses below.
>
> > **The authors claim in Remark 1 that the results in Theorems 1 and 2 are more general and unified compared with known results since they allow for any compact set $\mathcal{U}$**
>
> Thanks for this insightful comment. However, the error bound is usually more informative when a point is close to the solution set, as it characterizes the relationship between the distance to the complex solution set and the computable residuals. Ideally, a useful error bound should always be defined on a region containing an optimal solution so that when the computable residuals go to zero, we can deduce the distance to the solution is also zero. Our result for any compact set $\mathcal{U}$ containing an optimal solution is particularly useful in estimating the iteration number for local convergence. For example, suppose we have an algorithm that has a sublinear convergence rate of $\mathcal{O}(1/\epsilon)$ for any convex functions and linear convergence $\mathcal{O}(\log(1/\epsilon))$ for convex functions satisfying quadratic growth. Given the optimization problem, $\min_x f(x)$ with $f$ being convex and satisfying quadratic growth in the sublevel set $f_\nu = \\{ x \mid f(x) \leq \min_x f(x)   + \nu \\} $  with $\nu > 0$. In this case, one can infer the number of iterations that guarantee the iterate to reach the sublevel set $f_\nu$ using the $\mathcal{O}(1/\epsilon)$ convergence rate and claim the local linear convergence after iterates are within $f_\nu$.
>
> > **Aftrer Definition 2, the authors claim that the notion of strict complementarity is not restrictive, and it holds generically for conic programs. After that, the authors also presented: that it has been revealed that many structured SDPs from practical applications also satisfy strict complementarity. There are contradictions and misunderstandings in these two assertions.**
>
> Thanks for pointing out our contradicting statements. We will rewrite the paragraph to avoid misunderstandings in our updated version. For the statement regarding the generic property of conic programs, we mean that strict complementarity holds for almost all (in the sense of the Lebesgue measure) linear conic problems. For the statement regarding the structured SDPs, we mean that strict complementarity has been shown to hold for almost all SDPs with a specific structure, such as SDP relaxation of Max-cut and Matrix completion.
>
> > **Lines 184-185: If the linear constraints $\mathcal{A}(x) = b$ is empty, how to define the exact penalty function?**
>
> Thanks for this nice question. If the linear constraint $\mathcal{A}(x) = b$ is empty, then any penalty parameter will work for the penalty function for the conic constraint as the problem is infeasible.
>
> > **Line 226: Why is there $\mathbb{F} \times \mathbb{E}$ instead of \mathbb{F}?**
>
> Here, we follow our definition of the optimal solution set (1b). We write the dual solution set into the two spaces, so, in Line 226, $\mathbb{F}$ denotes the space where $y$ lives, and $\mathbb{E}$ denotes the space where $c-\mathcal{A}^*(y)$ lives.
>
> > **Please consider citing the paper "Rockafellar, R.T. Convergence of Augmented Lagrangian Methods in Extensions Beyond Nonlinear Programming. Math. Program. 199, 375–420 (2023). https://doi.org/10.1007/s10107-022-01832-5," and compare it with your work.**
>
> This paper discusses a more general setting of ALM, including nonconvex problems and convergence to a local minimum. However, as their setting is more general, the linear convergence of the primal variables also requires a strong assumption about the uniqueness of the primal and dual solutions.
>
> > **Typo: Line 238: Should (8b) be replaced by (8a)?**
>
> Thank you. This is a typo. It should be replaced by (8a).
>
> > **Typo: The notation $\mathbb{R}^n$ should be changed to $\mathbb{R}^n_+$.**
>
> We will double-check our paper and fix the typo. Thanks for your careful reading of our work.

---

> > ### Comment · Reviewer_X2mJ · 2024-08-12
> >
> > Thanks to the authors' responses, I have no further questions and will keep my score. However, I agree with the other reviewers that the paper's presentation could be improved.

---

### Official Review · Reviewer_Vd1D · 2024-07-13

**Soundness:** 3
**Presentation:** 3
**Contribution:** 3
**Rating:** 7
**Confidence:** 3

**Summary:**

This paper develops Inexact Augmented Lagrangian methods (ALM) for conic optimization problems with focus on Linear Programming (LP) or the non-negative orthant cone, Second-Order Cone (SOCP) problems and Semidefinite programming (SDP) given a linear objective. Under assumptions of strict complementarity and strong duality, the authors show local linear convergence for these problems by first showing quadratic growth property within a compact subset around the optimal solution for both the primal and dual programs and then leveraging the well-established concept that running ALM for the primal is equivalent to running Proximal Point Method (PPM) for the dual. The experiments show linear convergence for the SDP relaxation for the MaxCut problem, the SDP relaxation for matrix completion, and the moment relaxation of binary quadratic program (BQP) given the following metrics: primal distance, dual cost gap and the KKT-based residuals.

**Strengths:**

The biggest strength of the paper is the theoretical contribution to understanding of local linear convergence of inexact ALM methods for a variety of conic problems. Previous works have only shown local linear convergence of dual iterates but this work shows local  linear convergence for both primal and dual iterates. The authors provide a thorough theoretical discussion and provide all the proofs in the appendix section.

**Weaknesses:**

One weakness of the paper is the lack of characterization of how inexact evaluation of the proximal operator affects convergence. The authors use the property of linear convergence for Proximal Point Method (PPM) given inexact evaluation of proximal operator and quadratic growth assumption to show similar linear convergence for Inexact ALMs but there are no guarantees based on chosen error tolerance schedule (i.e. choice of $\epsilon_k$ and $\delta_k$ for all k). Another weakness of the paper is the lack of experimental evaluation of conic problems other than SDPs. The work shows theoretical guarantees of local linear convergence under strict complementarity and strong duality for three different types of conic problems: LPs, SOCPs and SDPs but experimental evaluation of LPs and SOCPs are lacking. It maybe because SDPs are a more general class than LPs or SOCPs but it would be good to have a variety of experiments that cover both convex/non-convex constraints as well as different LPs and SOCPs in addition to SDPs. There is also a lack of experimental evaluation to claim that there is local linear convergence given inexact evaluations of the subproblem in Equation (15a).

**Questions:**

1. Why were LPs and SOCPs not evaluated as part of the experimental evaluation but discussed in the theoretical sections and appendix?
2. Have you considered showing the local linear convergence given different random starting points for Inexact ALM?
3. Did you use inexact evaluation of the subproblem in Equation (15a)?
4. Have you considered showing the local linear convergence given different error tolerance schedules for $\epsilon_k$ and $\delta_k$ for all k?
5. Have you considered choosing the $\mathcal{A}$ operator to hold non-convex constraints in the experimental evaluation? The MaxCut SDP and Matrix Completion SDP both have linear constraints and the BQM has a convex constraint of $x_i^2 = 1$ but your guarantees only only assume that $\mathcal{A}$ is surjective.

**Limitations:**

There were no discussions of limitations in the work or adequate discussion of future directions of work.

---

> ### Author Rebuttal · Authors · 2024-08-06
>
> We appreciate the reviewer's time and effort in evaluating our manuscript. Your comments helped further improve the quality of our work. We provide detailed responses below
>
> > **Lack of characterization of how inexact evaluation of the proximal operator affects convergence.**
>
> Thanks for the valuable comment. It will be interesting to see how the inexactness ($\epsilon_k$ and $\delta_k$) propagates to the guarantee on the primal cost value gap, affine feasibility, and conic feasibility, similar to the work [1]. Our analysis considers the ALM as the dual side of the Proximal Point Method (PPM). Despite the lack of an analysis of the error propagation, the analysis based on PPM is simpler and more intuitive. We will incorporate the discussion on this part into the updated version of our paper.
>
> [1] Xu, Yangyang. "Iteration complexity of inexact augmented Lagrangian methods for constrained convex programming." Mathematical Programming 185 (2021): 199-244.
>
> >**Lack of experimental evaluation of conic problems other than SDPs.**
>
> Motivated by this comment, we have added more numerical experiments other than SDPs in the attached PDF file under the section of Auther rebuttal, l. The results also support our theoretical contributions.
>
> In particular, we consider two classical ML applications: linear SVM and Lasso. Both the linear SVM and Lasso can be reformulated as standard QP or SOCP. For each application, we randomly generate three instances and run ALM for those three instances with various augmented penalty parameters $\rho>0$. The numerical results show that the three quantities (primal cost value gap, dual cost value gap, and the KKT residuals) all converge to zero with a linear rate. The convergence speed of the primal feasibility and conic constraints also depends on the value of the penalty parameter $\rho$. The larger the penalty parameter, the faster the convergence of the feasibilities. We believe the oscillated or flattening behavior that happened in the tail (when the iterates are close to the solution set) is due to the inaccuracy of the subproblem solver and computational errors.
>
> [43] Ding and udell. On the simplicity and conditioning of low rank semidefinite programs. SIAM Journal on Optimization, 31(4):2614–2637, 2021.
>
> >**Lack of experimental evaluation to claim that there is local linear convergence given inexact evaluations of the subproblem**
>
> The linear convergence does happen under the inexactness (see conditions A' and B’ in the paper). This has been empirically verified in  SDPNAL+ [20]. Instead of directly dealing with the non-implementable conditions A’ and B’ (as A’ and B’ require the knowledge of the true cost value), they propose other alternative and implementable conditions and incorporate them into their subproblem solver.
>
> As our main focus is to establish the theoretical convergence guarantee for the primal iterates, we use the package YALMIP to formulate the subproblem and call MOSEK to solve the subproblem. It IS difficult to control the subproblem solution quality using a third-party package (MOSEK), as the subproblem is also solved to $\epsilon$-accurately by the conic solver. This is also revealed in our additional numerical experiments in the attached PDF. Specifically, the numerical experiments show some flattening behaviours in the tail when the iterates are close to the optimal solution set. We believe this is due to the computational error and the inexact subproblem solution.
>
> [20] Liuqin Yang, et al. Sdpnal+: a majorized semismooth Newton-CG augmented Lagrangian method for semidefinite programming with nonnegative constraints. Math. Program. Computation, 331–366, 2015.
>
> >**Have you considered showing the local linear convergence given different random starting points for Inexact ALM?**
>
> Throughout our numerical experiments, different random starting points all lead to the same convergence behavior as guaranteed by our theoretical result Theorem 3 which works for any initial points.
>
> >**Did you use inexact evaluation of the subproblem in Eq. (15a)?**
>
> We use the package YALMIP to formulate the subproblem and call the conic solver MOSEK to solve the subproblem. Therefore, the subproblem is solved to $\epsilon$-accuracy by the conic solver. Note that we can not guarantee the $\epsilon$ goes to zero as the ALM proceeds.
>
> >**Have you considered showing the local linear convergence given different error tolerance schedules for $\epsilon_k$ and $\delta_k$ for all $k$?**
>
> We did not investigate this part, as our main focus is the theoretical guarantee. We believe this will be more important in the algorithm development for solving the subproblem.
>
> >**Have you considered choosing the operator $\mathcal{A}$ to hold non-convex constraints in the experimental evaluation? The BQM has a convex constraint of $𝑥_𝑖^2=1$ but your guarantees only assume that $\mathcal{A}$ is surjective.**
>
> We did not solve the nonconvex BQM directly. Instead, we apply the moment relaxation with degree two to turn the nonconvex BQM into an SDP relaxation [61], and then solve the resulting SDP.
>
> [61] Jean B Lasserre. Global optimization with polynomials and the problem of moments. SIAM  Journal on optimization, 11(3):796–817, 2001.
>
> >**Limitations**
>
> Thanks for pointing out this issue in our writing. Our paper first establishes the growth property and error bound for three commonly used self-dual cones. Building upon the established growth property and error bound, our paper established the convergence rate of the primal iterates. The current theoretical guarantees do not apply to other cones. It will be interesting to see how to use the current proof techniques to establish the growth property and error bound for other non-symmetric cones. In addition, as solving the subproblem is the main step in the ALM, it will be interesting to develop an efficient subproblem tailored to the problem structures, such as sparsity or manifold structure.

---

> > ### Comment · Reviewer_Vd1D · 2024-08-12
> >
> > I would like to thank the authors for their detailed responses to my comments and questions. It is good to see the additional experimental results for linear SVM and Lasso, as this addresses the main concern about inadequate experimental evaluation, particularly with LPs and SOCPs. As far as showing how the error tolerance schedule $\epsilon_k$ and $\delta_k$ propagates to the guarantees, it is still a missing part of the analysis and useful to consider since achieving low error tolerance is difficult and/or costly in some applications. It is good that you will consider adding that analysis. I understand it is difficult to control the subproblem quality using a third-party solver like MOSEK but there are ways to set them. See this as an example for MOSEK: https://docs.mosek.com/latest/cxxfusion/parameters.html. Still, the additional experimental results merits an increase in my rating from 5 to 6.

---

> ### Author Response · Authors · 2024-08-13
> **Subproblem solution quality in Mosek**
>
> The authors would like to thank the reviewer for positive feedback and for directing us to the MOSEK documentation. We appreciate the reviewer's understanding of the challenges involved in conducting numerical experiments with varying schedules for $\epsilon_k$ and $\delta_k$.
>
> - Following the reviewer's suggestion, we conducted new numerical simulations to assess the impact of subproblem accuracy by varying the accuracy settings of the MOSEK solver.
> - We ran the inexact ALM on one instance of SVM and one instance of LASSO, adjusting the MOSEK accuracy settings (primal feasibility, dual feasibility, and relative complementarity gap) from $10^{-2}$ to $10^{-8}$.
> - Since we cannot include figures in this response, we present the numerical results in the following tables. The results indicate that higher accuracy in the subproblem solver leads to faster convergence of the residuals. For instance, with an accuracy setting of $10^{-2}$, the inexact ALM fails to converge to a high-accuracy solution after 20 iterations. With an accuracy setting of $10^{-4}$, the residuals converge to the order of $10^{-3}$. At $10^{-6}$ for the subproblem accuracy, the residuals decrease to the order of $10^{-6}$, and at $10^{-8}$, they decrease to around 4.62$\mathrm{e}$-08 after 20 iterations, showing linear convergence as expected by our theoretical analysis.
> - A similar effect of subproblem accuracy was observed in the LASSO experiments.
>
> Some detailed numerical results are shown below. We will add the corresponding figures to the final version of our work. We believe these extra numerical experiments further validate our theoretical analysis.
>
> > # **SVM - Primal cost value gap**
>
> | **Accuracy/Iteration** | **1**          | **5**          | **10**         | **15**         | **20**         |
> |------------------------|----------------|----------------|----------------|----------------|----------------|
> | **1.00$\mathrm{e}$-02** | 1.78$\mathrm{e}$-01 | 2.48$\mathrm{e}$-01 | 2.48$\mathrm{e}$-01 | 2.48$\mathrm{e}$-01 | 2.48$\mathrm{e}$-01 |
> | **1.00$\mathrm{e}$-04** | 4.29$\mathrm{e}$-01 | 1.58$\mathrm{e}$-02 | 6.07$\mathrm{e}$-03 | 5.98$\mathrm{e}$-03 | 5.97$\mathrm{e}$-03 |
> | **1.00$\mathrm{e}$-06** | 4.36$\mathrm{e}$-01 | 1.08$\mathrm{e}$-02 | 1.01$\mathrm{e}$-04 | 2.81$\mathrm{e}$-06 | 1.48$\mathrm{e}$-06 |
> | **1.00$\mathrm{e}$-08** | 4.37$\mathrm{e}$-01 | 1.07$\mathrm{e}$-02 | 2.70$\mathrm{e}$-04 | 7.29$\mathrm{e}$-06 | 4.62$\mathrm{e}$-08 |
>
>
> > # **Lasso - Primal cost value gap**
>
> | **Accuracy/Iteration** | **1**          | **5**          | **10**         | **15**         | **20**         |
> |------------------------|----------------|----------------|----------------|----------------|----------------|
> | **1.00$\mathrm{e}$-02** | 6.65$\mathrm{e}$+00 | 2.46$\mathrm{e}$-02 | 6.36$\mathrm{e}$-03 | 1.26$\mathrm{e}$-01 | 1.24$\mathrm{e}$-01 |
> | **1.00$\mathrm{e}$-04** | 6.78$\mathrm{e}$+00 | 1.13$\mathrm{e}$-03 | 1.27$\mathrm{e}$-03 | 3.37$\mathrm{e}$-03 | 2.14$\mathrm{e}$-03 |
> | **1.00$\mathrm{e}$-06** | 6.78$\mathrm{e}$+00 | 3.56$\mathrm{e}$-03 | 5.45$\mathrm{e}$-07 | 4.97$\mathrm{e}$-05 | 4.90$\mathrm{e}$-05 |
> | **1.00$\mathrm{e}$-08** | 6.78$\mathrm{e}$+00 | 3.50$\mathrm{e}$-03 | 4.97$\mathrm{e}$-05 | 6.89$\mathrm{e}$-07 | 1.06$\mathrm{e}$-08 |

---

### Official Review · Reviewer_K87V · 2024-07-13

**Soundness:** 3
**Presentation:** 3
**Contribution:** 3
**Rating:** 6
**Confidence:** 4

**Summary:**

The paper addresses the convergence of primal and dual iterates in Augmented Lagrangian Methods (ALMs) for conic optimization, particularly under quadratic growth assumptions. The authors establish that both primal and dual iterates of ALMs demonstrate linear convergence solely based on the strict complementarity assumption and a bounded solution set. This addresses a significant gap in the literature concerning the linear convergence of primal iterates, providing a theoretical foundation that has been previously unresolved.

**Strengths:**

Originality: The paper attempts to establish the conditions under which both primal and dual iterates of Augmented Lagrangian Methods (ALMs) for conic optimization exhibit linear convergence.

Clarity: The paper is well-structured and written with a clear narrative that guides the reader through the complex theoretical developments. Definitions, theorems, and proofs are delineated, making the complex content accessible to readers unfamiliar with the subject area.

Significance: The findings have potential implications for both the theoretical and practical aspects of optimization.

**Weaknesses:**

Presentation of Key Contributions: One significant issue with the paper is the placement and treatment of one of its key contributions, Theorem 3, which appears towards the very end of the document. This theorem, which is central to the paper's claims and theoretical framework, is not proven within the main text, and the appendix provided does not sufficiently validate or elaborate on it. This placement and lack of rigorous proof within the main discourse may detract from the impact of the contribution, as it might be overlooked or undervalued by readers. To improve, the authors could consider integrating Theorem 3 more prominently within the main discussion and ensuring that a complete and accessible proof is included either in the main text or more comprehensively in the appendix.

Numerical Validation: Another area where the paper falls short is in its numerical section. The section dedicated to numerical validation of the theoretical results is very brief and lacks depth/sufficient validation through several diverse problems. A more extensive set of numerical experiments is crucial to demonstrate the practical effectiveness and robustness of the proposed methods under varied conditions. The current numerical validation does not sufficiently cover the scope of the potential applications discussed in the paper. Readers are referred to Appendix H, where additional testing is still missing.

**Questions:**

Theorem 3 Proof: Can the authors provide a detailed proof of Theorem 3 within the main text or an enhanced appendix to help readers understand its foundational role in the paper's theoretical framework?

Numerical Experiments: Could the authors expand on the range and depth of the numerical experiments presented? Specifically, how do the proposed methods perform under varying conditions and parameter settings?

**Limitations:**

The paper does not explicitly discuss the limitations or potential negative societal impacts of the research, which is a critical component of scholarly communication.

---

> ### Author Rebuttal · Authors · 2024-08-06
>
> The authors are very thankful to the reviewer for taking the time and effort to review our manuscript. We sincerely appreciate all your valuable comments. All your comments are carefully addressed below.
>
> We have provided [some general responses](https://openreview.net/forum?id=Sj8G020ADl&noteId=WyA13MOOMr) in the Author Rebuttal by Authors above to two of your major concerns: 1) Presentation of Key Contributions; 2) Numerical Validation. In the following, we provide more specific responses to your comments on Theorem 3 Proof and Numerical Experiments.
>
> > **Theorem 3 Proof: Can the authors provide a detailed proof of Theorem 3 within the main text or an enhanced appendix to help readers understand its foundational role in the paper's theoretical framework?**
>
> Thanks for your nice suggestion. We will revise the introduction and incorporate further discussions about Theorem 3. Currently, Theorem 3 has two parts (“Dual iterates and KKT residuals” and “Primal iterates”). The first part “Dual iterates and KKT residuals” can be considered a primal version of [27, Theorem 1]. Due to the page limit, we provide the proof for the first part in Appendix E.2. The second part “Primal iterates” is one of our main contributions, and the proof relies on the error bound property for the primal conic problem established in Theorem 1 - (7b). In the updated version, we will highlight the key steps of the proof in the main text and provide a comprehensive and self-contained proof in the appendix.
>
> Moreover, we would like to point out that we have two main contributions to this work
>
> -  **Problem Structures:** We establish new quadratic growth and error bound properties for primal and dual conic programs (Theorem 1 and Theorem 2).
> - **Algorithm Analysis:** Utilizing our error bound properties, we prove that both primal and dual iterates of the ALMs converge linearly under mild assumptions (Theorem 3).
>
> Thus, Theorem 3 represents only one of the main theoretical contributions. The quadratic growth properties in Theorems 1 and 2 are equally important, and their proofs may be of independent interest (as noted in Remark 1 of the manuscript). The lack of presentation of Theorem 3 in the main content may be due to the page limit. To enhance the readability of the paper and better present Theorem 3,  we will follow the reviewers' suggestions and focus on the general problem class—SDP—in the main manuscript. This will not only improve clarity but also give us more space to ensure that Theorem 3 is well-integrated into the main discussion.
>
> [27] Ying Cui, Defeng Sun, and Kim-Chuan Toh. On the R-superlinear convergence of the KKT residuals generated by the augmented Lagrangian method for convex composite conic programming. Mathematical Programming, 178:381–415, 2019.
>
> > **Numerical Experiments: Could the authors expand on the range and depth of the numerical experiments presented? Specifically, how do the proposed methods perform under varying conditions and parameter settings?**
>
> Thank you for this suggestion. We have added more numerical experiments in the attached PDF file under the section of Auther rebuttal, and the results also support our theoretical contributions.
>
> In particular, we consider two classical machine learning applications: linear SVM and Lasso. Both the linear SVM and Lasso can be reformulated as standard conic programs with free, nonnegative, and second-order cone variables. For each application, we randomly generate three instances and run ALM for those three instances with various augmented penalty parameters $\rho>0$. The numerical results show that the three quantities (primal cost value gap, dual cost value gap, and the KKT residuals) all converge to zero with a linear rate. The convergence speed of the primal feasibility and conic constraints also depends on the value of the penalty parameter $\rho$. The larger the penalty parameter $\rho$, the faster the convergence of the residuals. The oscillated or flattening behavior that happens in the tail (when the iterates are close to the solution set) is due to the inaccuracy of the subproblem solver and computational errors.
>
> One key difference between the experiments on semidefinite programs (SDP) and the added experiments is the report of primal solution quality. In the considered SDP applications, “the SDP relaxation of maximum cut (Max-Cut) problem” and “the SDP relaxation of matrix completion” have been shown to have a unique primal solution with high probability [43]. Therefore, we can report the primal distance for SDP applications. On the contrary, the uniqueness of the primal solution for linear SVM and Lasso is not guaranteed, so we report the primal cost value gap instead.
>
> [43] Lijun Ding and Madeleine Udell. On the simplicity and conditioning of low rank semidefinite programs. SIAM Journal on Optimization, 31(4):2614–2637, 2021.
>
> > **Limitations: The paper does not explicitly discuss the limitations or potential negative societal impacts of the research, which is a critical component of scholarly communication.**
>
> Thanks for pointing out this issue in our writing. Our paper focuses on theoretical analysis of the structures of conic programs and linear convergence of ALM. We do not expect negative societal impacts in these aspects.
>
> Regarding limitations, our paper first establishes the growth property and error bound for three commonly used self-dual cones. Building upon the established growth property and error bound, our paper established the convergence rate of the primal iterates. The current theoretical guarantees do not apply to other cones. It will be interesting to see how to use the current proof techniques to establish the growth property and error bound for other non-symmetric cones. In addition, as solving the subproblem is the main step in the ALM, it will be interesting to develop an efficient subproblem tailored to the problem structures, such as sparsity or manifold structure.

---

> > ### Comment · Area_Chair_HRoC · 2024-08-14
> >
> > Dear Reviewer K87V,
> >
> > Are you satisfied with the rebuttal? Did the authors adequately address your concerns?
> >
> > Best,
> > AC

---

> > > ### Comment · Reviewer_K87V · 2024-08-14
> > >
> > > I thank the authors for making the effort to address the comments. I would be happy to increase the score to 6.

---

### Author Rebuttal · Authors · 2024-08-05

The authors would like to thank the four reviewers for their valuable time reading our manuscript and providing helpful comments. While the scores from the reviewers are mixed (3, 5, 8, 3), we find that their comments are generally positive and some are very constructive. In particular, all four reviewers appreciated the soundness of our work (all rated as good), and three reviewers (K87V, Vd1D and X2mJ) rated the contributions of our work as good or excellent.

Two major concerns raised by reviewers (K87V, Vd1D and rMRz) are
- **Presentation of one main theoretical result** (the linear convergence of inexact ALM in Theorem 3). Two reviewers (K87V and rMRz) pointed out that one key contribution in Theorem 3  appears very late in the paper  (Page 7). Reviewer K87V commented that the proof for Theorem 3 in the main text is not easy to follow. Reviewer rMRz also suggested that we could use a particular problem class (LP, SOCP, or SDP) to improve paper clarity.
- **Lack of sufficient numerical validations**. Reviewers K87V, Vd1D and rMRz requested more in-depth numerical validations using problems from a diverse range of applications. Reviewer Vd1D specifically suggested that experimental evaluations of LPs and SOCPs would be beneficial in validating the theoretical results.

In addition, Reviewer X2mJ provided many detailed comments which are useful to clarify our contributions further (the comment on Remark 1 and the suggestion of Rockafellar’s recent paper are very appreciated). All the four reviewers’s comments helped us further improve the quality of our work.

Here, we would like to respond to the two major concerns mentioned above (all other comments from the four reviewers have been carefully addressed below).

**Regarding the paper presentation**, we would like to emphasize that the contributions of this work have two key aspects:
1. *Problem Structures*: We establish new quadratic growth and error bound properties for primal and dual conic programs (**Theorem 1** and **Theorem 2**).
2. *Algorithm Analysis*: Utilizing our error bound properties, we prove that both primal and dual iterates of the inexact ALMs converge linearly under mild assumptions (**Theorem 3**).

Thus, Theorem 3 represents only one of the main theoretical contributions. The quadratic growth properties in Theorems 1 and 2 are equally important, and their proofs may be of independent interest (as noted in Remark 1 of the manuscript). In the paper presentation, we have attempted to balance these contributions. However, we agree with the reviewers that it would be beneficial to integrate Theorem 3 more prominently within the main discussion and provide a more complete and accessible proof in the appendix. In the updated version, we will follow the reviewers' suggestions and focus on the general problem class — SDP — in the main manuscript. This change will not only improve clarity but also give us more space to ensure that Theorem 3 is well-integrated into the main discussion. We will revise the proof of Theorem 3 to make it more accessible in the appendix.

**Regarding the numerical validations**: as the reviewers correctly pointed out, the major contributions of this work are theoretical. Due to the page limit, we have provided experiments on three problem classes (Max-Cut, matrix completion, binary quadratic program) to validate the practical convergence performance of ALM. We agree with the reviewers that more in-depth numerical validations will further demonstrate our theoretical contributions. To address this, we have conducted new experiments on two classical machine learning applications—linear SVM and Lasso. The results also supported our theoretical findings, and they are summarized in the attached PDF. In the final version, we will incorporate these new numerical results in the appendix.

We hope that the changes and responses will be sufficient to address the reviewers’ concerns. Please let us know if you have any additional questions or require further clarification. Any feedback will be highly appreciated.

---

### Decision · Program_Chairs · 2024-09-25

**Decision:**

Accept (poster)

**Comment:**

This paper introduces Inexact Augmented Lagrangian Methods (ALM) for addressing conic linear optimization problems. Under the assumptions of strict complementarity and strong duality, the authors establish local linear convergence by demonstrating the quadratic growth property within a compact neighborhood of the optimal solution for both the primal and dual problems.

I believe that the paper makes a significant contribution by proving the quadratic growth and error bounds for both primal and dual conic programs (P) and (D) over any compact set containing an optimal solution under standard assumptions, thereby extending existing results. Furthermore, the authors demonstrate that both primal and dual iterates converge linearly, a result that was previously unavailable for primal iterates. Most reviewers support the publication of the paper following the rebuttal. I recommend acceptance.